METHODS AND RESOURCES

# Genome-wide analysis of DNA replication and DNA double-strand breaks using TrAEL-seq

**Neesha Kara**[1], **Felix Krueger**[2], **Peter Rugg-Gunn**[1], **Jonathan Houseley**[1]*

**1** Epigenetics Programme, Babraham Institute, Cambridge, United Kingdom, **2** Babraham Bioinformatics, Babraham Institute, Cambridge, United Kingdom

* jon.houseley@babraham.ac.uk

**Data Availability Statement:** All sequencing files are available from the GEO database (accession number(s) GSE154811. Numerical data is in S1–S7 Data and image in S1 Raw Images.

## Abstract

Faithful replication of the entire genome requires replication forks to copy large contiguous tracts of DNA, and sites of persistent replication fork stalling present a major threat to genome stability. Understanding the distribution of sites at which replication forks stall, and the ensuing fork processing events, requires genome-wide methods that profile replication fork position and the formation of recombinogenic DNA ends. Here, we describe Transferase-Activated End Ligation sequencing (TrAEL-seq), a method that captures single-stranded DNA 3′ ends genome-wide and with base pair resolution. TrAEL-seq labels both DNA breaks and replication forks, providing genome-wide maps of replication fork progression and fork stalling sites in yeast and mammalian cells. Replication maps are similar to those obtained by Okazaki fragment sequencing; however, TrAEL-seq is performed on asynchronous populations of wild-type cells without incorporation of labels, cell sorting, or biochemical purification of replication intermediates, rendering TrAEL-seq far simpler and more widely applicable than existing replication fork direction profiling methods. The specificity of TrAEL-seq for DNA 3′ ends also allows accurate detection of double-strand break sites after the initiation of DNA end resection, which we demonstrate by genome-wide mapping of meiotic double-strand break hotspots in a *dmc1*Δ mutant that is competent for end resection but not strand invasion. Overall, TrAEL-seq provides a flexible and robust methodology with high sensitivity and resolution for studying DNA replication and repair, which will be of significant use in determining mechanisms of genome instability.

## Introduction

DNA double-strand breaks (DSBs) can be caused by exogenous agents (e.g., ionising radiation), defective cellular processes (e.g., replication–transcription collisions or topoisomerase dysfunction), or intentionally by the cell (e.g., in meiosis or immunoglobulin recombination) [1–3]. We have a detailed understanding of DSB repair pathways based on decades of research [4–6] but much less understanding of which pathways are used in a given genomic context in response to particular types of damage.

Prior to the introduction of high-throughput sequencing methods, genome-wide studies of DSB formation and processing were largely restricted to meiotic recombination, where

**Funding:** JH was funded by the Wellcome Trust [110216], JH, PRG and FK by the BBSRC [BI Epigenetics ISP: BBS/E/B/000C0423], NK was funded by the MRC [iCASE studentship] and Artios Pharma. The funders had no role in study design, data collection and analysis, decision to publish, or preparation of the manuscript.

**Competing interests:** The authors have declared that no competing interests exist.

**Abbreviations:** ARS, autonomously replicating sequence; DSB, double-strand break; hESC, human embryonic stem cell; RFB, replication fork barrier; rRNA, ribosomal RNA; TdT, terminal deoxynucleotidyl transferase; TrAEL-seq, Transferase-Activated End Ligation sequencing; TSS, transcriptional start site; UMI, unique molecular identifier.

frequent DSBs at well-defined sites can be stabilised either before or after end resection and mapped on microarrays [7–9]. However, these microarray methods lacked the signal-to-noise ratio required for DSB detection in other situations, and so the development of the direct DSB sequencing method BLESS marked a step change in mapping technologies [10]. In BLESS, an adaptor is directly ligated to the DSB end to prime Illumina sequencing reads, allowing precise mapping and relative quantification of breaks. Modifications of BLESS have improved ligation efficiency (END-seq [11], DSB-capture [12]), quantitation (qDSB-seq [13], BLISS [14]), signal-to-noise and generality (BLISS [14], i-BLESS [15]), and variants have been developed for specific systems including meiosis (S1-seq [16]). These methods differ in detail but all involve blunting of the DNA end with nuclease activities that remove 3′ extended single-stranded DNA to form a double-stranded end for adaptor ligation. This can be a problem as end resection forms long tracts of 3′ extended single-stranded DNA each side of a DSB that are degraded by blunting, such that the sequencing adaptor is ligated to the chromosomal DNA many kilobases from the original break site if resection has occurred. Other strategies for DSB mapping include direct labelling of DNA ends with biotin or extracting protein-linked DNA on glass fibre, to allow fragment purification prior to ligation of sequencing adaptors (Break-seq, CC-seq) [17–19]; however, like BLESS, these yield the locations of 5′ rather than 3′ ends. Therefore, if resection has occurred, the original location of DNA breaks as opposed to the end point of end resection cannot be mapped by any of these methods, which is problematic as DSB repair is often easiest to inhibit postresection (such as in classic *rad51Δ* or *rad52Δ* mutants in yeast).

Profiles yielded by DSB mapping methods can rarely be considered in isolation as replication has a dramatic influence on the distribution of DNA strand breaks in a cell [13,15]; replication defects can be a primary cause of DNA damage but replication also provides both opportunity and the requirement to repair existing lesions. Replication forks moving rapidly through chromosomes stall at protein obstacles, DNA damage, and through collisions with the transcription machinery [20–22], and must be restarted by pathways that carry an increased risk of mutation [20–23]. Understanding the distribution and causes of DNA damage across the genome therefore requires integration of DSB profiles with approaches to monitor DNA replication.

Many methods for mapping DNA replication have been developed, which can be broadly divided into those which measure copy number changes through S-phase and those which analyse replication forks or replication bubbles directly. Copy number analysis stratifies the genome based on replication timing and defines early and late-firing origins [24–27]. This requires segregation of cell populations at different stages of replication or between replicating and non-replicating cells, either by cell cycle synchronisation or, more flexibly, by flow cytometry. Copy number methods are well refined, and the innate simplicity of this approach has even allowed application to single cells, revealing surprising uniformity in replication profiles across mammalian cells [28,29]. However, these methods do not have the resolution to detect individual origins in mammalian cells unless markedly different in timing, and a range of other more specialised approaches have been applied to study replication initiation [30,31], particularly by isolating short nascent DNA strands to identify individual origins or initiation zones [32–34]. Methods have also been developed to detect replication fork directionality through isolation and sequencing of Okazaki fragments (OK-seq) [35,36]; as well as revealing origins, these methods identify regions that are uniformly replicated in the forward or reverse direction and termination zones in which replication direction will vary depending on the point at which forks converge in individual cells. Although powerful, methods for direct analysis of forks and origins are technically demanding since replication bubbles, short nascent strands and Okazaki fragments are rare species that need to be carefully separated from each

other and from contaminating genomic DNA. As an alternative, PU-seq uses a relatively simple DNA library preparation to identify leading and lagging strands based on ribonucleotide incorporation but does require very specific DNA polymerase mutants with reduced ribonucleotide discrimination [37].

Direct ligation of a sequencing adaptor to the 3′ end of individual DNA strands would be a very attractive means of quantifying DNA damage irrespective of DNA resection, and direct labelling of DNA 3′ ends may reveal replication fork direction, particularly in mutants unable to ligate Okazaki fragments. Some methods aimed at mapping single-strand breaks and base changes theoretically have this capability [38,39], and very recently, the Ulrich lab described such a method, GLOE-seq, that is capable of replication profiling in DNA ligase-deficient yeast and human cells and also maps DSBs, although activity on resected substrates was not tested [40]. Here, we describe an alternative method, Transferase-Activated End Ligation sequencing (TrAEL-seq), which accurately maps DNA 3′ ends at DSBs that have undergone DNA resection. Remarkably, in addition to resected DSBs, we find that TrAEL-seq can profile DNA replication fork direction with excellent sensitivity even in wild-type yeast and mammalian cell populations without labelling or synchronisation.

## Results

### Implementation of TrAEL-seq

Various ligases can attach single-stranded DNA linkers to the 3′ end of single-stranded DNA, but efficiency is generally poor. An alternative method described by Miura and colleagues utilises terminal deoxynucleotidyl transferase (TdT) to add 1 to 4 adenosine nucleotides onto single-stranded DNA 3′ ends, forming a substrate for DNA adaptor ligation by RNA ligases [41,42] (Fig 1A steps i and ii). On a test substrate in vitro, TdT added 1 to 3 nucleotide A tails to >95% of single-stranded DNA molecules, which was ligated with approximately 10% efficiency to TrAEL-seq adaptor 1 using truncated T4 RNA ligase 2 KQ (Fig 1B).

TrAEL-seq adaptor 1 is a hairpin that primes conversion of single-stranded ligation products to double-stranded DNA suitable for library construction, incorporates a biotin moiety flanked by deoxyuracil residues that allows selective purification and elution of ligation products, and includes an 8-nucleotide unique molecular identifier (UMI) for bioinformatic removal of PCR duplicates (Fig 1A). Once TrAEL-seq adaptor 1 is ligated, a thermophilic polymerase with strong strand displacement and reverse transcriptase activities extends the hairpin to form unnicked double-stranded DNA (Fig 1A, step iii), then the DNA is fragmented by sonication and adaptor-ligated material is purified on streptavidin magnetic beads (Fig 1A, steps iv and v). The DNA ends formed during fragmentation are polished and ligated to TrAEL adaptor 2 while still attached to the beads (Fig 1A, step vi), then the purified fragments flanked by TrAEL adaptors 1 and 2 are eluted by cleavage of the deoxyuracil residues prior to library amplification (Fig 1A, step vii). The resulting library is sequenced using a primer that anneals to TrAEL-seq adaptor 1, such that **the TrAEL-seq read is the reverse complement of the original DNA 3′ end** (Fig 1A, step viii).

### Detection of 3′ extended DNA ends by TrAEL-seq

We tested TrAEL-seq on agarose-embedded yeast genomic DNA digested with restriction enzymes *Not*I, *Pme*I, and *Sfi*I that yield 5′ extended, blunt, and 3′ extended ends, respectively, and generated a BLESS-type END-seq library from the same digested material for comparison (Fig 1C). The resulting TrAEL-seq library contained fragments of 200 to 2,000 bp as expected (S1A Fig), and sequencing data was processed through a custom bioinformatic pipeline to remove the A-tail, map the reads, and deduplicate by UMI (illustrated in S1B Fig). Comparing

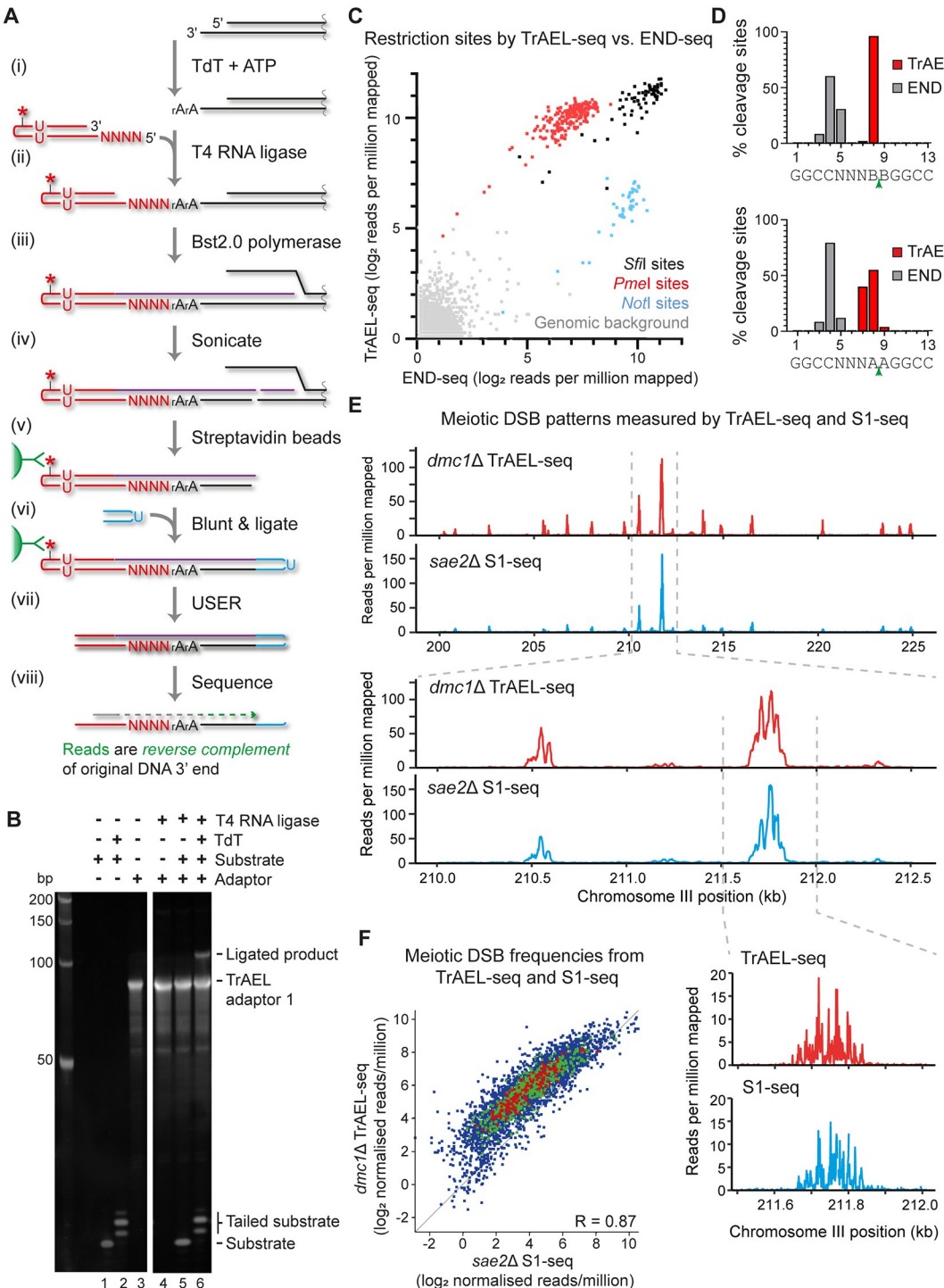

**Fig 1. TrAEL-seq accurately maps and quantifies 3′ ends of DNA. (A)** Schematic representation of the TrAEL-seq method. Agarose-embedded genomic DNA is used as a starting material, plugs are washed extensively to remove unligated TrAEL adaptor 1, and agarose is removed prior to Bst 2.0 polymerase step. The blunting and ligation of TrAEL-adaptor 2 is performed using a NEBNext Ultra II DNA kit, and TrAEL-adaptor 2 homodimers removed by washing streptavidin beads before USER enzyme treatment. The finished material is ready for PCR amplification using the NEBNext amplification system. Note that TrAEL-seq reads map antisense to the cleaved strand, reading the complementary sequence starting from the first nucleotide before the cleavage site. *—biotin moiety, U—deoxyuracil, N—any DNA base, rA—adenosine. **(B)** In vitro assay of adaptor ligation. An 18-nucleotide single-stranded DNA oligonucleotide was treated with or without TdT, then ligated to TrAEL adaptor 1 using T4 RNA ligase 2 truncated KQ. Products were separated on a 15% PAGE gel and visualised by SYBR Gold

staining. (**C**) Scatter plot comparing read counts from yeast DNA digested with *Sfi*I, *Pme*I, and *Not*I, along with the genome average, based on END-seq and TrAEL-seq. Note that the genome average signal encompasses all single-copy 13 bp regions that do not overlap with a site, while restriction enzyme quantitation represents reads mapping to 13 bp around the recognition site (*Sfi*I site is 13 bp, *Not*I / *Pme*I sites were extended to 13 bp). (**D**) Precision mapping of *Sfi*I cleavage sites by TrAEL-seq and END-seq. *Sfi*I sites, which contain 5 degenerate bases were split into those that contain no A's at the cleavage site (GGCCNNNB|BGGCC, 87 sites, upper panel) or A's flanking the cleavage site (GGCCNNNA|AGGCC, 15 sites, lower panel), considering cleavage sites on forward and reverse strands separately. Mapped locations of 3′ ends were averaged across each category of site and expressed as a percentage of all 3′ ends mapped by each method to that category of site. (**E**) Comparison of meiotic DSB profiles from *dmc1*Δ cells performed by TrAEL-seq and *sae2*Δ cells by S1-seq (SRA accession: SRP261135) [45]. Both techniques should map Spo11 cleavage sites in the given mutants. Regions of 25 kb and 2.5 kb on chromosome III are shown for reads counted in 20 bp windows. The lowest panel shows 500 bp around the major peak for reads counted at single bp resolution. (**F**) Scatter plot of log-transformed normalised read counts at all 3,907 Spo11 cleavage hotspots annotated by Mohibullah and Keeney, comparing *dmc1*Δ TrAEL-seq with *sae2*Δ S1-seq data (right) [16,45,47] (SRA accession: SRP261135). Numerical data underlying this figure can be found in S1 Data, gel image in S1 Raw Images. DSB, double-strand break; TdT, terminal deoxynucleotidyl transferase; TrAEL-seq, Transferase-Activated End Ligation sequencing.

TrAEL-seq and END-seq data shows that both methods detect restriction enzyme cleavage sites: Efficiency is approximately equal on 3′ extended ends, END-seq is more efficient on 5′ extended ends, while TrAEL-seq unexpectedly performed better on the blunt *Pme*I ends (Fig 1C). Therefore, both methods efficiently detect DSBs even though the labelling strategies are very different.

The restriction enzyme *Sfi*I has a degenerate recognition sequence (GGCCNNNN| NGGCC) that allows assessment of TrAEL-seq ligation efficiency on different 3′ end sequences, allowing us to ensure that there is no bias for DNA ends based on the 3′ or adjacent nucleotides (S1C Fig). Fine mapping of cleavages at the *Sfi*I recognition site GGCCNNNN| NGGCC reveals differences between END-seq and TrAEL-seq: END-seq, in common with other BLESS-type methods, degrades the 3′ overhang and returns a consensus cleavage location 3′ of nucleotides 4 to 5 of the recognition site (Fig 1D). In contrast, TrAEL-seq can map the real cleavage site (3′ of nucleotide 8) and does so for >98% events, but only for *Sfi*I sites lacking A nucleotides adjacent to the cleavage site (i.e., GGCCNNNB|BGGCC) (Fig 1D, top). This problem stems from the A-tails added by TdT, which cannot be distinguished from genome-encoded A's. To reconcile this issue, we used a trimming algorithm that removes up to a maximum of 3 T's from the start of the read. Since the average tail length is 2 to 4 nucleotides, this correctly maps the *Sfi*I cleavage site to nucleotides 7 to 9 in >98% of reads, even when only the most challenging sites for mapping are considered (those with the structure GGCCNNNA|AGGCC) (Fig 1D, bottom). Importantly, this algorithm does not overtrim ends within genome-encoded A tracts such that the 10 *Sfi*I sites with 2 or more 3′ A's (GGCCNNAA|NGGCC) are mapped with the same accuracy (S1D Fig). We suggest that this overall mapping accuracy of >98% within ±1 nucleotide would be sufficient for almost all applications.

A major strength of TrAEL-seq should be the ability to map original sites of DSBs even after resection, a point in the homologous recombination process that is particularly amenable to stabilisation using mutations that prevent strand invasion. We chose meiosis as an in vivo model system to validate this as meiotic DSB patterns have been extremely well characterised. Meiotic DSBs formed by Spo11 are processed by Sae2 among other factors prior to resection, after which strand invasion into a homologous chromosome is mediated by Dmc1 [43,44]. Loss of Sae2 therefore stabilises DSBs prior to resection, whereas loss of Dmc1 stabilises DSBs after resection and before strand invasion. TrAEL-seq for the 3′ ends of resected DSBs in *dmc1*Δ cells 7 h after induction of meiosis revealed a DSB pattern very similar to that observed for unresected DSBs in an *sae2*Δ mutant mapped by S1-seq (a BLESS variant specific for meiotic recombination) (Fig 1E) [45]. TrAEL-seq technical replicates are highly reproducible

across known hotspots of Spo11 cleavage (R = 0.99) (S1E Fig), and quantitation of these hotspots by TrAEL-seq correlates well to S1-seq in *sae2Δ* cells (R = 0.87) (Fig 1F, left) and Spo11 oligonucleotide sequencing (R = 0.85) (S1F Fig) [46,47]. Of the 3,907 known hotspots, TrAEL-seq detects 3,542 based on a threshold of 2 SDs above background, which lies between S1-seq (2,556), and Spo11 oligonucleotide sequencing (a much more labour-intensive method that forms the gold standard for meiotic DSB mapping, 3,784). TrAEL-seq sensitivity is broadly similar to CC-seq (a method specialised for protein-associated DNA ends [19]), which detects 3,223 sites by the same criteria. This shows that TrAEL-seq accurately maps and quantifies endogenous DSB sites even after end resection. Importantly, meiotic recombination is unusual in that mutants are known which completely stabilise DSBs, whereas stabilising breaks postresection is often more practical in other systems.

Overall, TrAEL-seq provides an effective method for detecting and quantifying DSBs genome-wide even after end resection.

## High-resolution mapping of stalled replication forks by TrAEL-seq

Replication forks stall at various impediments during DNA replication and stalled forks may undergo reversal or cleavage as the cell attempts to restart replication (Fig 2A). The replication fork barrier (RFB) in the rDNA of budding yeast is a classic system for studies of replication fork stalling, and results from replication forks encountering the Fob1 protein bound to DNA [48]. Fob1 binds just downstream of the 35S ribosomal RNA (rRNA) gene and prevents the passage of replication forks moving against the direction of 35S transcription that would otherwise encounter the RNA polymerase I machinery head-on [49,50]. The RFB has been intensely studied as a model for stalled replication forks initiating recombination and genome rearrangement [51,52], and DSBs thought to stem from fork cleavage have been reported at the RFB based both on Southern blotting and qDSB-seq (a BLESS-type method for mapping double stranded DNA ends) [13,53,54].

To detect replication forks stalled at the RFB and test the requirement for homologous recombination in resolution of these species, we prepared TrAEL-seq libraries from unsynchronised wild-type, *fob1Δ*, and *rad52Δ* cells growing at mid-log phase: *fob1Δ* cells lack RFB activity, while *rad52Δ* mutants cannot initiate homologous recombination. RFB signals should therefore be absent from *fob1Δ*, while signals representing DSBs formed by fork cleavage should accumulate in *rad52Δ* as this mutant cannot repair such DNA breaks once formed.

Two RFB sites are clearly visible in wild-type TrAEL-seq data as peaks of reverse strand reads but are absent in the *fob1Δ* mutant (Fig 2B, wild type and *fob1Δ* panels). These peaks are exactly reproduced between 2 libraries prepared independently from the same fixed cells (by different investigators working 6 months apart, S2A Fig) and are detected with high signal-to-noise in 3 wild-type biological replicates (S2B Fig). These sites correspond well with the RFB sites mapped using high-resolution gels [53,55] and are also visible in published qDSB-seq and GLOE-seq datasets, although TrAEL-seq data contains fewer additional peaks in this region than GLOE-seq data and the RFB peaks correspond more closely to known sites than qDSB-seq peaks (Fig 2B) [13,40].

To determine the applicability of TrAEL-seq to mammalian cells, we generated 2 TrAEL-seq datasets each from 2 biological replicate libraries of 0.5 million human embryonic stem cells (hESCs). A major peak was observed in the rDNA downstream of the RNA polymerase I termination site in both hESC biological replicates, on the reverse strand located in the most distal of the known RFB sites (Fig 2C) [56]. This observation is consistent with an efficient polar RFB located just downstream of the RNA polymerase I transcription unit, as seen in diverse species from plants to yeast to mice [49,57–60]. Furthermore, we detect smaller but

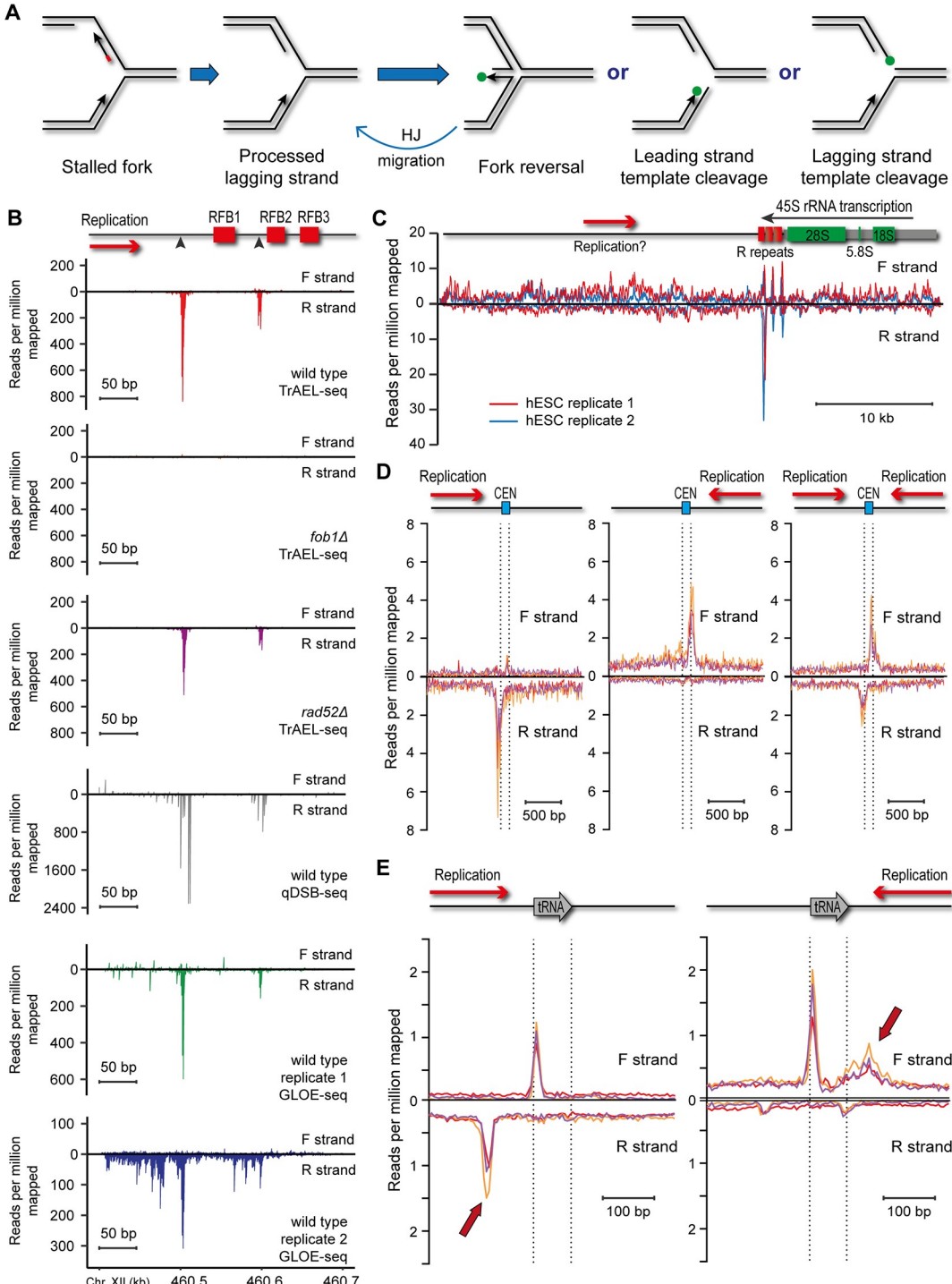

**Fig 2. Visualisation of replication fork stalling sites by TrAEL-seq.** (**A**) Potential processing pathways of a stalled replication fork. Lagging strand processing is likely to finish soon after stalling, and at least for the yeast RFB, it is known that the lagging strand RNA primer is removed [55]. The fork could then undergo fork reversal to yield a Holliday junction or be cleaved on the leading or lagging strand. Whereas cleavage is irreversible and requires a recombination event to restart the replication fork, reversed forks can revert to the normal replication fork structure by Holliday Junction migration (labelled HJ migration). The 3′ DNA ends predicted to be TrAEL-seq substrates are labelled with green dots. The RNA primer on the Okazaki fragment in the leftmost structure is shown in red. (**B**) Comparison of the yeast rDNA RFB signals in TrAEL-seq datasets compared to qDSB-seq (SRA accession: SRX5576747) [13] and GLOE-seq (SRA accessions: SRX6436839 and SRX6436840) [40]. Reads were quantified in 1 nucleotide steps and normalised to reads per million mapped. qDSB-seq data were obtained from S-phase

synchronised cells, all other samples are from asynchronous log-phase cell populations growing in YPD media. Schematic diagram shows the positions of RFB elements previously mapped by 2D gel electrophoresis [49,50], and black triangles indicate previously mapped sites of DNA ends [53,55]. (**C**) rDNA TrAEL-seq reads in hESCs. Two biological replicates are shown, each an average of 2 technical replicates. Reads were summed in 100 bp sliding windows spaced every 10 bp. One rDNA repeat is shown, the RNA polymerase I-transcribed 45S RNA is shown as a grey line with mature rRNAs marked in green in the schematic diagram. Note that the 45S gene is shown as transcribed right to left to maintain consistency with the yeast data, such that the sequence is the reverse complement of the rDNA reference sequence U13369. The R repeats, which contain the RFBs, are marked in green, while the primary direction of replication is shown by a red arrow labelled as "Replication?" to take into account evidence that forks can move in both directions through the human rDNA. (**D**) Average TrAEL-seq profiles across centromeres +/− 1 kb for 3 biological replicates of wild-type cells (drawn in red, orange, and purple). Centromeres are categorised based on replication direction in the yeast genome assembly into those replicated forward (CEN3, CEN5, CEN13, CEN2), reverse (CEN11, CEN15, CEN10, CEN8, CEN12, CEN9), and those in termination zones that could be replicated in either direction (CEN14, CEN16, CEN1, CEN4, CEN7, CEN6), see S2C Fig for details. Read counts per million reads mapped were calculated in nonoverlapping 10 bp bins, vertical lines indicate annotated boundaries of centromeres. (**E**) Average TrAEL-seq profiles across tRNAs +/− 200 bp for 3 biological replicates of wild-type cells (drawn in red, orange, and purple). tRNAs are categorised into those for which transcription is codirectional with the replication fork and those for which transcription is head-on to the direction of the replication fork. tRNAs for which the replication direction is not well defined were excluded. Arrows indicate peaks that are dependent on replication direction. Read counts per million reads mapped were calculated in nonoverlapping 5 bp bins, vertical lines indicate annotated boundaries of tRNAs. Numerical data underlying this figure can be found in S2 Data. hESC, human embryonic stem cell; RFB, replication fork barrier; rRNA, ribosomal RNA; TrAEL-seq, Transferase-Activated End Ligation sequencing.

reproducible peaks on both strands in all 3 RFB sites, consistent with the low efficiency bidirectional RFB activity that has been reported in human cells based on 2D gels and DNA combing (Fig 2C) [56,61,62].

rDNA RFBs are not the only sites at which replication forks stall, for example, reported GLOE-seq peaks at yeast centromeres likely stem from replication forks stalling at centromeric chromatin [40,63]. To probe this relationship, we first stratified centromeres into those replicated only by reverse forks, those replicated only by forward forks, and those sited in termination zones where forks converge (S2C Fig). At centromeres replicated from one direction only, we observed an accumulation of reads on the opposite strand to the direction of replication located just before the centromere, while forks in termination zones that can be replicated in either direction displayed both peaks (Fig 2D and S1 File). A similar analysis of tRNA loci, which are also known to stall replication forks [64], yielded more complex patterns (Fig 2E). These regions displayed peaks upstream or downstream of the tRNA depending on the direction of replication (Fig 2E, arrows), consistent with previous studies that reported both codirectional and head-on tRNA transcription can stall replication forks, at least in the absence of replicative helicases [64–67]. However, we also observed a major peak covering the first approximately 15 bp of the tRNA gene, which was not affected by replication direction and appears to mark a transcription-associated break on the template strand that must be a conserved feature of tRNA transcription as it is also detected in the hESC samples (S2D Fig). This aside, we find that sites of replication fork stalling both at the RFB and other sites are revealed by an accumulation of TrAEL-seq reads on the opposite strand to the direction of replication.

The structures resulting from stalled fork processing have various double-stranded 3′ ends that should be substrates for TrAEL-seq based on our restriction enzyme analysis (Figs 1C and 2A, green dots). However, no difference in signal intensity was observed between *rad52Δ* and wild type at the rDNA, centromeres or tRNAs, showing that these double-stranded ends are not normally processed by the homologous recombination machinery (Fig 2B, S2E and S2F Fig). DSBs formed in the rDNA are known to be repaired by homologous recombination, and although we and others have reported Rad52-independent recombination at the rDNA, these are rare events unknown in wild-type cells [68–70]. If TrAEL-seq peaks represented fork cleavage events, we would expect a strong stabilisation in the *rad52Δ* mutant. So, based on the lack of stabilisation observed, we consider that the vast majority of DNA ends at sites of replication

fork stalling represent reversed forks that can revert to normal replication fork structures by Holliday Junction migration without recombination (see Fig 2A and Discussion).

Taken together, these results show that TrAEL-seq allows sensitive and precise mapping of replication fork stalling, most likely through labelling of reversed replication forks.

## TrAEL-seq profiles describe replication fork directionality

A striking feature of yeast TrAEL-seq data is the massive variation in strand bias of reads at different sites in the genome: A violin plot of the fraction of reverse reads in 1 kb bins shows 2 distinct peaks at 15% to 30% and 70% to 85%, a behaviour much less obvious in comparable GLOE-seq data (Fig 3A) [40]. TrAEL-seq read polarity in asynchronous wild-type cells (calculated from the difference between reverse and forward read densities) forms clear domains when plotted over large genomic regions that almost perfectly match the GLOE-seq map of Okazaki fragment ends in a Cdc9 DNA ligase depletion experiment, although with the opposite polarity (Fig 3B and S3A Fig) [40]. Mapping of Okazaki fragment ends is a well-validated method for detecting replication forks [35,36], and the tight correlation of TrAEL-seq data to Okazaki fragment distribution strongly suggests that TrAEL-seq detects processive replication forks even in wild-type cells. Indeed, the locations at which TrAEL-seq polarity switches from negative to positive coincide precisely with replication origins (autonomously replicating sequence or ARS elements) (Fig 3B, dotted vertical lines), and alignment of TrAEL-seq reads across 30 kb either side of all ARS elements reveals a switch in polarity as would be expected for replication forks diverging from replication origins (Fig 3C). Furthermore, TrAEL-seq reads in the rDNA reflect the known role of Fob1 in enforcing unidirectional rDNA replication, as reads are highly polarised in wild-type cells but this polarisation is absent in *fob1Δ* (S3B Fig).

Absolute TrAEL-seq read density is largely uniform across the single-copy genome, except for pronounced dips at each ARS (Fig 3D), suggesting that TrAEL-seq signals are primarily derived from active replication forks with little underlying noise. If so, then TrAEL-seq signals should vary across the cell cycle. However, as with other sequencing methods, quantitative comparison of total TrAEL-seq signal between libraries is not straightforward, as there is no relationship between total read count in a library and amount of substrate in the original sample. To allow such comparisons, we modified the TrAEL-seq pipeline such that 2 samples are barcoded at an early stage and then pooled for processing, sequencing, and postprocessing as a single sample. This approach maintains the absolute ratio of substrate between the 2 samples, allowing quantitative comparison.

We applied this method to compare cells arrested in G1 using α-factor to cells from the same culture after release into S-phase. Two variants of TrAEL-seq adaptor 1 with unique barcodes were ligated to the G1 and G1->S samples which were then pooled, and in each experiment, we performed 2 technical replicates with the barcodes swapped to ensure that no quantitative differences emerged from the adaptors themselves. Two biological replicate experiments yielded essentially identical results, with the TrAEL-seq read count across single-copy regions being dramatically higher in the G1->S samples than in the G1-arrested samples. To illustrate both absolute read quantity and strand bias, we plotted the read counts on forward and reverse strands separately across chromosome V (Fig 3E); S-phase samples show strong signals that phase between forward and reverse reads across the chromosome, whereas signals from G1 cells are almost undetectable. Furthermore, the phasing between forward and reverse matches the read polarity variation of unsynchronised samples (compare Fig 3B and 3E). This experiment shows that TrAEL-seq signals primarily arise from active DNA replication forks and are very low in nonreplicating cells.

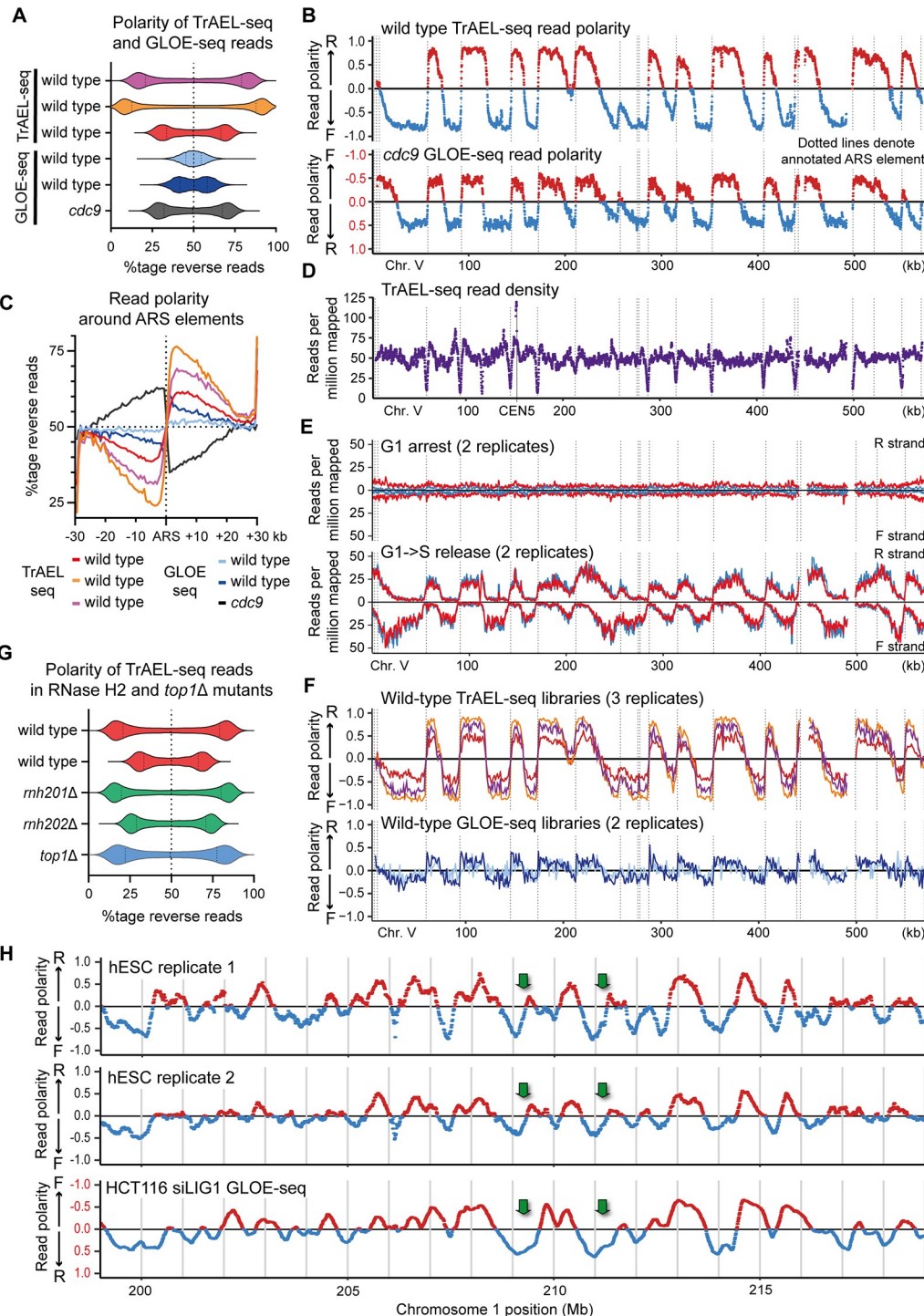

**Fig 3. TrAEL-seq is highly sensitive to replication fork direction. (A)** Polarity of TrAEL-seq and GLOE-seq reads assessed in 1 kb windows across the genome excluding windows overlapping multicopy regions, presented as the percentage of total reads that map to the reverse strand. The dotted line marks 50%, which equates to an absence of strand bias. TrAEL-seq libraries are 3 biological replicates of BY4741 wild type. GLOE-seq wild-type samples (SRA accessions: SRX6436839 and SRX6436840) were derived from asynchronous log phase cells growing in YPD, as were the TrAEL-seq samples. The *cdc9* dataset is of synchronised cells depleted of the DNA ligase Cdc9 (SRA accession: SRX6436838). **(B)** Read polarity plots for TrAEL-seq BY4741 wild type growing at log phase on YPD and GLOE-seq Cdc9 depletion data (SRA accession: SRX6436838) across chromosome V, calculated as (R−F)/(R+F) where R and F indicate reverse and forward reads, respectively. TrAEL-seq data are an average of 2 technical replicates. Read polarity was calculated for 1,000

bp sliding windows spaced every 100 bp for all single-copy regions; gaps near 450 kb and 500 kb are Ty elements. Vertical dotted lines show locations of ARS elements. Note that the read polarity axis of the *cdc9* data is inverted for easy comparison to TrAEL-seq as the *cdc9* mutation enriches for 3′ ends on the lagging strand, whereas TrAEL-seq detects the 3′ end of the leading strand. (**C**) Average read polarity of TrAEL-seq and GLOE-seq datasets across 30 kb windows either side of annotated ARS elements. Calculated as the %tage of reverse reads amongst all reads. Samples are as in **A**. (**D**) Absolute TrAEL-seq read depth in reads per million mapped irrespective of read polarity, for the same sample shown in **B**. Read depth is broadly uniform across the single-copy genome except for a peak at the centromere (as in Fig 2D) and dips at each active ARS. (**E**) TrAEL-seq signals in wild-type cells arrested in G1 (top) or released into S (bottom). Read counts per million reads mapped were calculated for 1,000 bp sliding windows spaced every 100 bp for all single-copy regions, and strands are shown separately to reveal both the absolute read count and the read polarity at each point—read polarity distribution across the chromosome for S-phase cells is equivalent to Fig 3B. To allow comparison of read counts between 2 samples, G1 and G1->S samples were ligated to TrAEL adaptor 1 variants carrying 2 different barcodes. These samples were then pooled, processed, and sequenced together to maintain the relative read counts between the samples, and normalisation for each sample was to the total reads mapped across both libraries. To ensure that the different adaptor barcodes did not impact the result, 2 technical replicates were performed for each paired sample of G1 and G1->S with the barcode adaptors inverted. Data shown are an average of the technical replicates, but little difference was observed in relative library quantification that could be attributed to barcoding. Two biological replicates for the experiment are shown in red and blue. (**F**) Strength and reproducibility of read polarity amongst TrAEL-seq and GLOE-seq datasets. Read polarity was calculated in 1,000 bp windows spaced every 1,000 bp and shown as continuous lines. Three biological replicate datasets for wild-type TrAEL-seq are plotted on the upper graph and show the same replication profiles. Two wild-type GLOE-seq datasets are overlaid on the lower graph (SRA accessions: SRX6436839 and SRX6436840). TrAEL-seq and GLOE-seq datasets all derive from asynchronous cultures harvested during log phase growth in YPD [40]. Vertical dotted lines show locations of ARS elements. (**G**) Read polarity plot as in **A** for 2 biological replicates of BY4741 wild-type TrAEL-seq datasets compared to the RNase H2 mutants *rnh201Δ* and *rnh202Δ* and to topoisomerase I mutant *top1Δ*. (**H**) Read polarity plots of TrAEL-seq data for asynchronous wild-type hESCs, 2 biological replicates are shown each an average of 2 technical replicates. GLOE-seq data of LIG1-depleted HCT116 cells (average of SRA accessions: SRX7704535 and SRX7704534) are shown for comparison. Read polarity was calculated in 250 kb sliding windows spaced every 10 kb. Note that the polarity of the HCT116 data has been inverted to aid comparison with TrAEL-seq samples; this is highlighted by the scale being labelled in red. Profiles are broadly similar between the 2 cell types, but some origins are only active in hESCs; examples are indicated by green arrows. Numerical data underlying this figure can be found in S3 Data. ARS, autonomously replicating sequence; hESC, human embryonic stem cell; TrAEL-seq, Transferase-Activated End Ligation sequencing.

Phasing of read polarity was also noted in wild-type samples profiled by GLOE-seq but only weakly, whereas TrAEL-seq libraries display very strong read polarity differences that are highly reproducible and yield essentially identical replication profiles (Fig 3A and 3E, S3C Fig) [40]. As Sriramachandran and colleagues noted for GLOE-seq [40], the read polarity of this replication signal is opposite to what would be expected from labelling of 3′ ends in normal forks. There should never be fewer 3′ ends on the lagging strand than the leading strand, yet up to 90% of TrAEL-seq reads emanate from the leading strand. To explain the GLOE-seq signal, Sriramachandran and colleagues suggested that GLOE-seq labels sites at which DNA is nicked during removal of misincorporated ribonucleotides [40]. To test this idea, we generated TrAEL-seq libraries from *rnh201Δ* and *rnh202Δ* mutants that lack key components of RNase H2, the main enzyme that cleaves DNA at misincorporated ribonucleotides, along with a wild-type control [71,72]. Strikingly, read polarity in these mutants is equivalent to wild type, showing that the leading strand bias of TrAEL-seq reads is not caused by RNase H2 and therefore is unlikely to arise through excision of misincorporated ribonucleotides (Fig 3G and S3D Fig). It is also possible that TrAEL-seq (and indeed GLOE-seq) signals arise when the replication machinery encounters Top1 cleavage complexes [73], but we saw no reduction in TrAEL-seq polarity or signal in *top1Δ* cells (Fig 3G and S3D Fig). One further observation in this regard is that END-seq data show a polarity bias, albeit weak, that parallels the polarity bias in TrAEL-seq data generated from the same cells (S3E Fig). This suggests that double-stranded ends are also formed during normal replication, although these faint signals could also arise through cleavage of the delicate single-stranded regions of replication forks during processing.

We then asked if an equivalent strand bias is observed in the hESC libraries. The limited read coverage in these libraries only allowed read polarity to be determined in 250 kb

windows, but nonetheless, a striking variation was observed across the genome (Fig 3H). Importantly, these profiles were very similar between technical and biological replicates and cannot therefore simply result from noise; this can be observed across defined genomic regions but is also clear in a scatter plot which shows that the average read polarity within each window correlates between the datasets (R = 0.84, S3F and S3G Fig). Furthermore, comparison to GLOE-seq results from a LIG1-depleted human cell line that is defective in Okazaki fragment ligation again revealed a striking similarity to the hESC TrAEL-seq data, although with the opposite polarity (Fig 3H and S3H Fig) [40]. Interestingly, a subset of origins were reproducibly detected in hESC samples but absent in the HCT116 data, consistent with evidence that origin usage differs between these cell lines (Fig 3H, green arrows) [24].

We therefore conclude that TrAEL-seq primarily detects processive replication forks and does so with exceptionally high signal-to-noise. TrAEL-seq profiles are highly reproducible and can be obtained from wild-type cells without need for cell synchronisation, sorting, or labelling. The 3′ ends detected by TrAEL-seq correspond to the leading rather than the lagging strand, despite the fact that many more 3′ ends occur on the lagging strand, and we suggest that these 3′ ends are exposed by replication fork reversal occurring either in vivo or during sample processing (see Discussion).

## Environmental impacts on replication timing and fork progression

Finally, we asked whether TrAEL-seq can reveal replication changes or DNA damage, and in particular whether we can detect collisions between transcription and replication machineries.

Since all the yeast libraries generated up to this point had yielded essentially identical DNA replication profiles outside the rDNA, we were first keen to ensure that changes in replication profile are indeed detectable. We therefore examined cells lacking Clb5, a yeast cyclin B that plays a key role in the activation of late-firing replication forks [74]. The TrAEL-seq profile of *clb5Δ* was very similar to wild type across most of the genome, but certain origins were clearly absent or strongly repressed, resulting in extended tracts of DNA synthesis from adjacent origins visible as regions of very different polarity (Fig 4A, green arrows, S4A Fig). This is as predicted for *clb5Δ* mutants and confirms that TrAEL-seq is indeed sensitive to changes in replication profile.

We then engineered collisions between RNA polymerase II and the replisome by changing growth conditions to strongly induce certain genes; specifically, we added galactose to cells growing on raffinose, which strongly induces expression of galactose metabolising genes including *GAL1*, *GAL7*, and *GAL10*. Although these genes are adjacent, *GAL1* is transcribed codirectionally with the replication fork, whereas *GAL7* and *GAL10* are orientated head-on to the fork (Fig 4B, schematic). On one hand, stalled replication forks have not been observed at this locus by 2D gels [65], but conversely, the strong activation of the *GAL1–10* promoter has proven highly recombinogenic in various assays [75–77]. We performed these experiments in wild-type cells and in a strain lacking both Dnl4, the DNA ligase required for nonhomologous end joining, and Rad51, the recA ortholog which mediates strand invasion for homologous recombination. *dnl4Δ rad51Δ* double mutants should be unable to repair DSBs irrespective of cell cycle phase and therefore should accumulate any DSBs that form.

Collisions would seem most likely where the replisome passes through the transcribed region of highly expressed genes oriented head-on to the direction of replication (such as *GAL10* or *GAL7*), so we predicted that any consequent replication fork stalling would occur at the 3′ end of the gene or within the open reading frame. However, TrAEL-seq read densities across the *GAL* gene cluster provided little evidence for transcription-associated replication fork stalling within gene bodies. Instead, peaks of reverse reads formed at the 5′ end of the

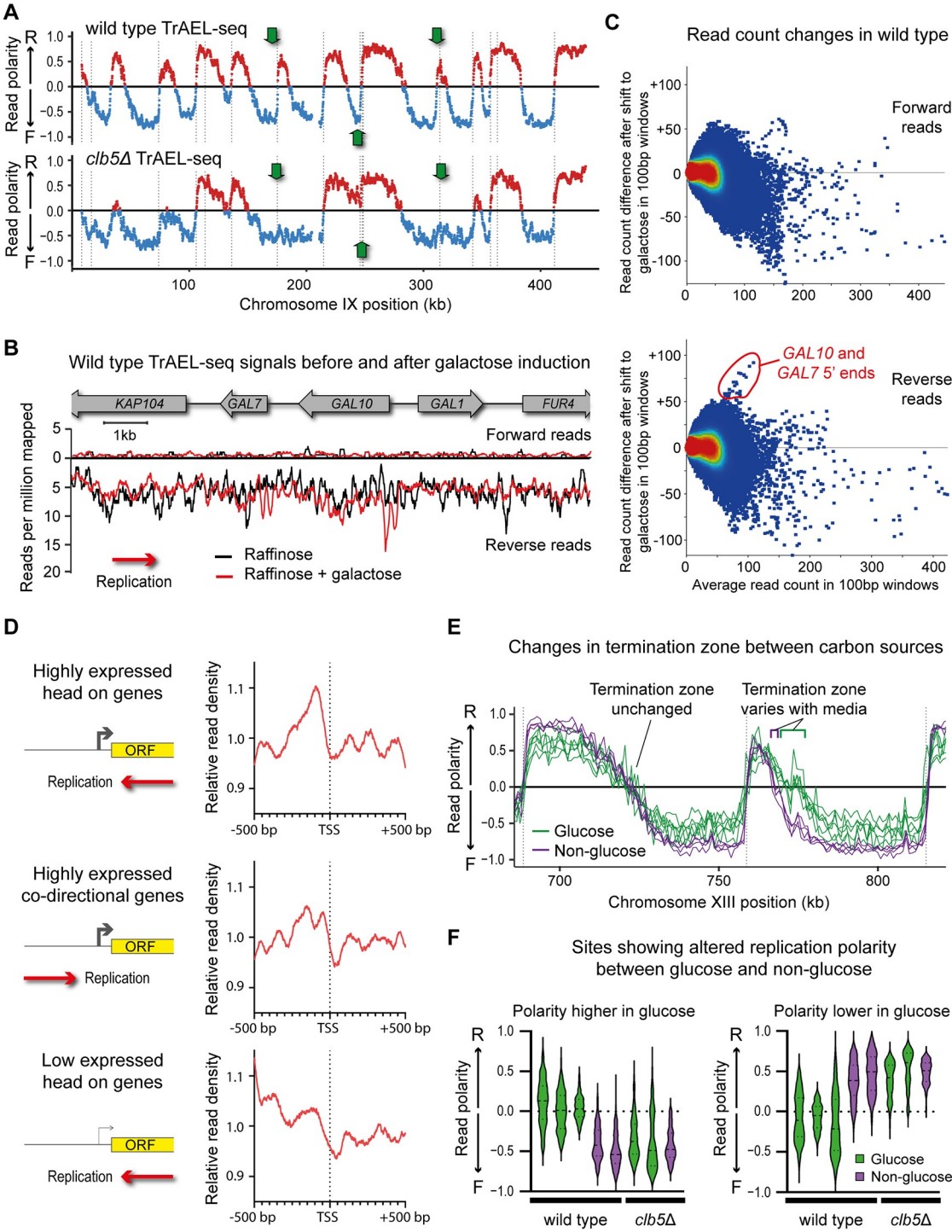

**Fig 4. Detection of replication variation using TrAEL-seq.** (**A**) Read polarity plot for TrAEL-seq data of *clb5Δ* versus wild type over a representative region of chr IX. Arrows indicate ARS elements that are not activated in the absence of Clb5. (**B**) Line plot showing forward and reverse strand TrAEL-seq read counts across the GAL genes for wild-type cells maintained on YP raffinose or 5 h after addition of galactose to 2%. Reads were quantified in 100 bp sliding windows spaced every 10 bp. (**C**) MA plots showing the change in read count against the average read count for each 100 bp window in the single-copy genome between cells maintained on raffinose and cells exposed to galactose. Separate plots are shown for forward and reverse reads; read counts were normalised to total library size. (**D**) Plots of average TrAEL-seq read density around the TSS in the highest or lowest 25% expressed genes based on NET-seq data for wild-type yeast growing on YPD (SRA: SRX031059). Genes were categorised into those orientated head-on or codirectional with replication based on TrAEL-seq replication profiles. Data are shown for wild-type BY4741 cells growing on YPD.

(**E**) Example location in which a termination zone differs depending on carbon source. Read polarity was calculated in 1 kb windows spaced every 1 kb. Green lines show cells grown on glucose and purple lines cells grown on raffinose or raffinose plus galactose. (**F**) Violin plots of regions showing large and significant read polarity differences between cells grown on glucose and nonglucose carbon sources (defined using sets given below). Read polarity data are shown for wild type and *clb5Δ* grown on glucose (green) and raffinose or raffinose plus galactose (purple). Differences observed in wild type are suppressed in *clb5Δ*. To define this set of regions, read polarity was calculated across the single-copy genome in 1 kb windows, then each window was compared between the 2 sets by *t* test with a Benjamini and Hochberg correction. As many samples as possible were included in these sets for best separation based on media: glucose (3 replicates of wild type plus *rad52Δ*, *rnh201Δ*, *rnh202Δ*) and nonglucose (wild type on raffinose, wild type on raffinose + galactose, *dnl4Δ rad51Δ* on raffinose, *dnl4Δ rad51Δ* on raffinose + galactose). Windows were then filtered for those with a difference in read polarity >0.4 between the 2 sets, leaving a set of 196 out of 12,182 (2.3%). Plots were split based on the direction of the difference in read polarity for clarity. Numerical data underlying this figure can be found in S7 Data. ARS, autonomously replicating sequence; TrAEL-seq, Transferase-Activated End Ligation sequencing; TSS, transcriptional start site.

*GAL10* gene, and also of the *GAL7* gene, although the latter was less prominent, which suggests that the replication fork is stalled by chromatin or proteins bound at the promoter after passing through the body of the gene (Fig 4B and S4B Fig). The read accumulation is not dramatic, but compared to the rest of the single-copy genome, these sites showed the largest increase in read count between cells on raffinose only and those on raffinose plus galactose (Fig 4C and S4C Fig). As for the sites of fork stalling described above, we detected little difference between the recombination defective mutant (*dnl4Δ rad51Δ*), and the wild type showing that promoter signals must represent fork stalling events that are rarely processed to recombinogenic DSBs (S4B and S4C Fig). Furthermore, the region in which replication forks passing through the *GAL* locus encounter oncoming forks from *ARS211* was unchanged on galactose, meaning that delays caused by fork stalling must be very transient (S4D Fig). Our evidence for minimal replisome pausing even at the most highly expressed genes contrasts with previous estimates based on DNA polymerase or γH2A occupancy [78,79] but is in keeping with more recent studies that have not observed defects in fork progression or activation of Mec1 when replication forks encounter highly transcribed genes [66,80].

To determine whether such signals are unique to the *GAL* genes, we categorised yeast genes both by orientation to the replication fork and by expression based on published NET-seq data for YPD [81] and derived plots of average TrAEL-seq read density around transcriptional start sites (TSS) for wild-type cells growing on YPD. Highly expressed genes (top 25% by NET-seq) orientated head-on to the replication fork show a small but sharp peak before the TSS (Fig 4D, top panel). This peak is dependent on replication, being absent from highly expressed genes orientated codirectionally with the replication fork, and also from highly expressed head-on genes in G1-arrested cells (Fig 4D, middle panel, S4E Fig). Similarly, the peak depends on transcription and is absent from head-on genes in the bottom 25% of expressed genes (Fig 4D, bottom panel). This shows that replication forks are more prone to pausing at the TSS of highly expressed head-on orientated genes; we also note that TrAEL-seq signals from these genes phase around the TSS with nucleosome spacing, suggesting these interactions reinforce nucleosome positioning.

Unexpectedly, we noted changes in termination zones elsewhere in the genome when comparing the 4 samples from the galactose induction experiment, which were grown on raffinose or raffinose with galactose, to other wild-type and mutant TrAEL-seq libraries for which cells were grown on glucose (see, for example, Fig 4E). Comparing cells based on growth media rather than genotype, we discovered significant and substantial ($p < 0.01$, average read polarity change >0.4) differences in read polarity for approximately 2% of the single-copy genome. The most prominent differences affected a subset of termination zones where the average site at which forks converge moved by up to 10 kb (Fig 4E). This change would be most easily attributed to a change in replication timing, and indeed the *clb5Δ* mutant, although grown on glucose, showed the same average read polarity at the media-dependent sites as the cells grown

on nonglucose carbon sources (raffinose and/or galactose) (Fig 4F). This suggests that the timing of replication firing is altered depending on carbon source, consistent with a previous report that Clb5 nuclear import is suppressed in yeast growing in ethanol [82].

Together, these data show that replication profiling by TrAEL-seq is sufficiently sensitive to reveal differences in fork direction and processivity.

## Discussion

Here, we have demonstrated that TrAEL-seq maps the 3′ ends of resected DSBs, sites of replication fork stalling and normal DNA replication patterns genome-wide and with base pair resolution. Methods to map the 3′ ends of resected DNA are desirable for genome-wide studies of homologous recombination as these are the critical species that undergo strand invasion. Similarly, detection of DNA 3′ ends at stalled replication forks is an important indicator of potentially recombinogenic intermediates. TrAEL-seq profiles all these species with excellent signal-to-noise and therefore provides a general method for the detection of DNA processing events that could result in genome instability. It is interesting to note that the primary source of noise in TrAEL-seq is actually normal replication forks. This raises questions as to the frequency with which leading strand 3′ ends become detached during normal replication (discussed below) but also provides a major unanticipated application for the method. In contrast to other methods for profiling replication fork directionality (notably through Okazaki fragment sequencing), TrAEL-seq works in wild-type cells, requires neither labelling nor synchronisation of cells, and does not involve complex sample preparation procedures, making TrAEL-seq versatile and straightforward to implement across a range of experimental contexts.

### A proposed mechanism for replication fork detection by TrAEL-seq

TrAEL-seq was designed to detect free 3′ ends of single-stranded DNA and was not expected to label undisturbed replication forks in normal cells. Why therefore is TrAEL-seq so sensitive to replication fork direction? Although TrAEL-seq may have some capacity to label 3′ ends in normal replication fork structures, we cannot see why TrAEL-seq would outperform GLOE-seq in detecting such ends, and the bias towards the leading strand would be very hard to explain. Instead, we suggest that replication forks frequently rearrange, either in vivo or during sample processing, to make the leading strand 3′ end accessible to TdT while the lagging strand 3′ end remains largely inaccessible. Transient fork reversal would have this effect, yielding TdT-accessible leading strand ends without irreversible changes in fork structure (Fig 5, free 3′

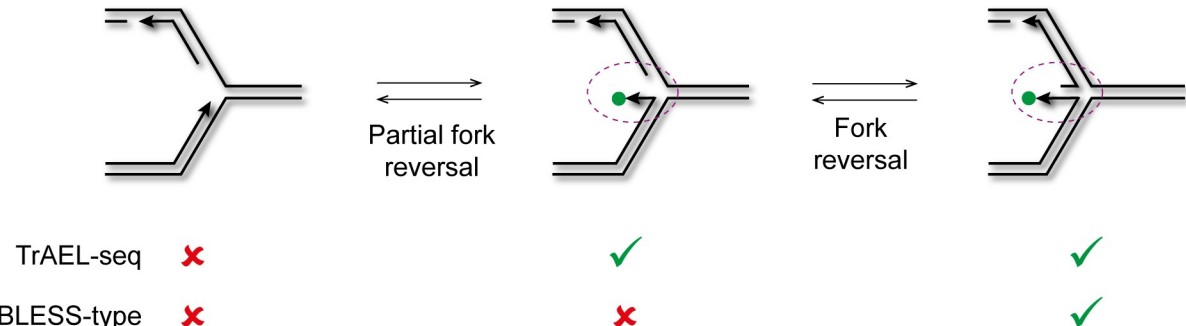

**Fig 5. Proposed mechanism for replication fork detection by TrAEL-seq.** Replication forks that would normally be undetectable by TrAEL-seq undergo very limited reversal to yield a free 3′ end that can be labelled by TdT (green dot, middle structure). Further reversal yields a double-stranded end that can be labelled by TrAEL-seq or BLESS-type methods. Purple circles highlight the area of difference between the structures. TdT, terminal deoxynucleotidyl transferase; TrAEL-seq, Transferase-Activated End Ligation sequencing.

ends labelled with green dots). Only a small subset of these events need to undergo sufficient reversal for the nascent lagging and leading strands to anneal, which would form the replication-linked double-stranded DNA ends that we detect by END-seq (Fig 5, middle and right structures, S3E Fig). It remains to be determined if these rearrangements occur in vivo, and if so would require surprisingly frequent fork reversal, although for TrAEL-seq labelling the reversal required is minimal—in reality only a flap displacement (Fig 5, left and middle structures). Although DNA replication is highly processive overall, in vitro measurements have shown that the yeast leading and lagging strand polymerases dissociate after less than 1 kb of DNA synthesis [83], and this may allow helicases to access and unwind the nascent leading strand.

Alternatively, it is possible that the TrAEL-seq replication signal derives from cleaved replication forks, but we think this is highly unlikely for the following reasons: (1) The *rad52Δ* mutant used here had almost no growth defect and showed no detectable difference in TrAEL-seq profile, and (2) there is no difference in detection of early and late replicating genome regions in TrAEL-seq, whereas the activity of structure-specific endonucleases that could cleave replication forks is tightly restricted to G2/M [84]. Replication-linked double-stranded DNA ends have been clearly observed by BLESS-type methods in cells exposed to replication stress [13,15,85] and interpreted as evidence that replication forks are cleaved either during the restart process or as a pathogenic end point. However, fork cleavage is not required to initiate recombination during replication fork restart [86], and it is quite possible that apparent DSBs are actually double-stranded ends of reversed forks. Direct observation of cleaved forks at the rDNA RFB has been reported based on Southern blot [53,54,68], but we note that these signals could also arise from fork reversal (S5 Fig). This distinction is important as cleaved forks must be resolved by recombination of some sort, whereas reversed forks can revert by Holliday Junction migration. Overall, the existence of frequent DSBs in wild-type cells under normal conditions (quantified at 1 DSB per cell per S-phase for the RFB alone [13]) is hard to reconcile with the minimal growth phenotype of mutants lacking critical DNA repair factors such as Rad52. We suggest that the vast majority of such events detected by TrAEL-seq and other DNA end-mapping methods are actually reversed replication forks that are rapidly resolved by fork migration.

## Complementary methods probe different aspects of DNA damage

Although TrAEL-seq and the recently described GLOE-seq method in theory act equivalently by labelling and profiling DNA 3′ ends, we find that these methods have completely different strengths and weaknesses. TrAEL-seq proves superior for detection of replication fork direction and stalling, which likely arises through a sensitivity to replication fork structure. In contrast, the DNA denaturing step required for GLOE-seq labelling erases fork structure and reveals real accumulations of strand breaks as opposed to conformational changes in the replication fork. Therefore, future studies employing both methods in parallel are likely to be particularly informative for understanding the dynamics of replication forks on encountering obstacles. It should also be noted that the lack of a denaturing step in TrAEL-seq makes it insensitive to single-strand breaks and nicks, and therefore GLOE-seq is much better suited for detection of such ends.

Genome-wide analysis of DNA processing events requires high-resolution methods that can detect changes at both 5′ and 3′ DNA ends. BLESS-type methods degrade or fill in 3′ ends to yield the location of matching 5′ ends, and our implementation of TrAEL-seq now provides a complementary method to map 3′ ends. We suggest that for dissecting mechanisms of DSB processing and repair, these methods will be most powerful when employed together. In

addition to the TrAEL-seq protocol, we therefore also provide an implementation of BLESS/ END-seq that utilises small numbers of cells and follows the same library construction procedure as TrAEL-seq, making processing of the same sample in parallel by both methods straightforward. Indeed, we have successfully performed TrAEL-seq and END-seq on two-halves of the same agarose plug.

For general replication analysis, most existing methods profile either fork direction or origin timing, whereas acquisition of information on both parameters from the same samples would be very helpful. The recently described D-Nascent method can determine fork direction and origin timing, but only after cell synchronisation and label incorporation [87]. The ability of TrAEL-seq to obtain replication direction profiles from asynchronous unlabelled wild-type cells will allow easy integration with other methods under diverse growth conditions. For example, ethanol fixed cells collected for sort-seq [27] could also be profiled by TrAEL-seq to provide both replication timing and direction. However, some adjustments will be needed when combining TrAEL-seq with replication timing methods that involve labelling with deoxyuridine derivatives (e.g., REPLI-seq) as USER is employed in TrAEL-seq to elute libraries prior to amplification.

Overall, TrAEL-seq provides a unique addition to complement existing methods for genome-wide analysis of DNA replication and DNA damage. The relatively simple experimental protocol, high signal-to-noise ratio, and lack of requirement for treatment or purification of cells prior to harvest should render TrAEL-seq particularly suitable for a wide range of experimental systems.

## Materials and methods

### Yeast strains and culture

Strains used are listed in S1 Table. All media components were purchased from Formedium, all media was filter sterilised. YP media was supplemented with the given carbon source from 20% filter-sterilised stock solutions. For growth to log phase, cells were inoculated in 4 ml media and grown for approximately 6 h at 30°C with shaking at 200 rpm before dilution at approximately 1:10,000 in 25 ml YPD (1:500 for YP raffinose or 1:2,000 for synthetic complete media) and growth continued at 30°C 200 rpm for approximately 18 h until OD reached 0.4 to 0.7 (mid-log). Cells were centrifuged 1 min at 4,600 rpm, resuspended in 70% ethanol at $1 \times 10^7$ cells/ml and stored at −70°C.

For meiosis, SK1 $dmc1\Delta$ diploid cells from a glycerol stock were patched overnight on YP 2% Glycerol then again for 7 h on YP 4% glucose before inoculating in 4 ml YPD and growth for 24 h, then inoculated to OD 0.2 in 20 ml YP acetate for overnight growth to approximately $4 \times 10^7$ cells/ml in a 100-ml flask at 30°C with shaking at 200 rpm. Meiosis was initiated by washing cells once with 20 ml SPO media (0.3% KOAc, 5 mg/L uracil, 5 mg/L histidine, 25 mg/L leucine, 12.5 mg/L tryptophan, 0.02% raffinose), then resuspending in 20 ml SPO media and incubating for 7 h at 30°C in a 100-ml flask with shaking at 250 rpm. Cells were harvested and fixed with 70% ethanol as above.

For G1 arrest, BY4741 wild-type cells were grown in 20 ml YPD at 30°C 200 rpm for approximately 18 h to $0.5 \times 10^7$ cells/ml (mid-log), then α-factor added to 5 μg/ml (from Zymo Y1001 stock diluted to 5 mg/ml in DMSO) and cells maintained at 30°C 200 rpm for 1 h. Another aliquot of α-factor was added to 10 μg/ml total and cells maintained at 30°C 200 rpm for 1 more hour. At this point, >90% cells were Schmoos and no small budded cells were visible. Half the cells were harvested by centrifugation 1 min at 4,600 rpm and resuspended in 70% ethanol at $1 \times 10^7$ cells/ml. The other half were centrifuged 1 min at 4,600 rpm, washed twice with prewarmed YPD at 30°C, then resuspended in 10 ml prewarmed YPD and

transferred to a prewarmed 25 ml flask. Cells were maintained at 30°C 200 rpm until most cells showed small buds (approximately 50 min), then harvested as above. All cells were stored at −70°C.

## hESC culture

Undifferentiated H9 hESCs were maintained on Vitronectin-coated plates (ThermoFisher Scientific A14700) in TeSR-E8 media (StemCell Technologies 05990). All hESCs were cultured in 5% $O_2$, 5% $CO_2$ at 37°C.

## Agarose embedding of yeast cells

Cells in ethanol (1 to $3 \times 10^7$ per plug) were pelleted in round bottom 2 ml tubes by centrifuging 30 s 20,000g, washed once in 1 ml PFGE wash buffer (10 mM Tris HCl (pH 7.5), 50 mM EDTA) and resuspended in 60 μl same with 1 μl lyticase (17 U/ μl in 10 mM $KPO_4$ pH7, 50% glycerol, Merck >2,000 U/mg L2524). Samples were heated to 50°C for 1 to 10 min before addition of 40 μl molten CleanCut agarose (Bio-Rad 1703594), vortexing vigorously for 5 s before pipetting in plug mould (Bio-Rad 1703713) and solidifying 15 to 30 min at 4°C. Each plug was transferred to a 2-ml tube containing 500 μl PFGE wash buffer with 10 μl 17 U/μl lyticase and incubated 1 h at 37°C. Solution was replaced with 500 μl PK buffer (100 mM EDTA (pH 8), 0.2% sodium deoxycholate, 1% sodium N-lauroyl sarcosine, 1 mg/ml Proteinase K) and incubated overnight at 50°C. Plugs were rinsed with 1 ml TE, then washed 3 times with 1 ml TE for 1 to 2 h at room temperature with rocking; 10 mM PMSF was added to the second and third washes from 100 mM stock (Merck 93482). Plugs were then digested 1 h at 37°C with 1 μl 1,000 U/ml RNase T1 (Thermo EN0541) in 200 μl TE. RNase A was not used as it binds strongly to single-stranded DNA [88]. Plugs were stored in 1 ml TE at 4°C and are stable for >1 year.

## Agarose embedding of hESC cells

Cells were detached using Accutase, counted and $1 \times 10^6$ cells were washed once in 5 ml L buffer (10 mM Tris HCl (pH 7.5), 100 mM EDTA, 20 mM NaCl) and resuspended in 60 μl L buffer in a 2-ml tube. Samples were heated to 50°C for 2 to 3 min before addition of 40 μl molten CleanCut agarose (Bio-Rad 1703594), vortexing vigorously for 5 s before pipetting in plug mould (Bio-Rad 1703713), and solidifying 15 to 30 min at 4°C. Each plug was transferred to a 2-ml tube containing 500 μl digestion buffer (10 mM Tris HCl (pH 7.5), 100 mM EDTA, 20 mM NaCl, 1% sodium N-lauroyl sarcosine, 0.1 mg/ml Proteinase K) and incubated overnight at 50°C. Plugs were washed and RNase T1 treated as for yeast.

## TrAEL-seq library preparation and sequencing

Please note that a detailed TrAEL-seq protocol is provided in S2 File, and up-to-date protocols are available from the Houseley lab website https://www.babraham.ac.uk/our-research/epigenetics/jon-houseley/protocols

Preparation of TrAEL-seq adaptor 1: DNA oligonucleotide was synthesised and PAGE purified by Sigma-Genosys (Merck, United Kingdom):

[Phos]NNNNNNNNNAGATCGGAAGAGCGTCGTGTAGGGAAAGAGTGTUGCGCAG GCCATTGGCC[BtndT]GCGCUACACTCTTTCCCTACACGACGCT

This oligonucleotide was adenylated using the 5′ DNA adenylation kit (NEB, E2610S) as follows: 500 pMol DNA oligonucleotide, 5 μl 10× 5′ DNA adenylation reaction buffer, 5 μl 1 mM ATP, 5 μl Mth RNA ligase in a total volume of 50 μl was incubated for 1 h at 65°C then 5

min at 85˚C. Reaction was extracted with phenol:chloroform (pH 8), then ethanol precipitated with 10 µl 3M NaOAc, 1 µl GlycoBlue (Thermo AM9515), 330 µl ethanol and resuspended in 50 µl 0.1x TE.

Preparation of TrAEL-seq adaptor 2: DNA oligonucleotide was synthesised and PAGE purified by Sigma-Genosys (Merck):

[Phos]GATCGGAAGAGCACACGTCTGAACTCCAGTCUUUUGACTGGAGTTCAGA CGTGTGCTCTTCCGATC*T

Oligonucleotide was annealed before use: 20 µl 100 pM/µl oligonucleotide and 20 µl 10x T4 DNA ligase buffer (NEB) in 200 µl final volume were incubated in a heating block 95˚C 5 min, then block was removed from heat and left to cool to room temperature over approximately 2 h.

Sample preparation: ½ an agarose plug was used for each library (cut with a razor blade), hereafter referred to as a plug for simplicity. All incubations were performed in 2 ml round bottomed tubes (plugs break easily in 1.5 ml tubes), or 15 ml tubes for high volume washes. For restriction enzyme digestion, a plug was equilibrated 30 min in 200 µl 1x CutSmart buffer (NEB), digested overnight at 37˚C with 1 µl 20 U/µl *Not*I-HF (NEB R3189S) and 1 µl 10 U/µl *Pme*I (NEB R0560S) in 400 µl 1x CutSmart buffer, then 1 µl 20 U/µl *Sfi*I (NEB R0123S) was added and incubation continued overnight at 50˚C. The plug was rinsed with 1x TE before further processing.

Tailing and ligation: Plugs were equilibrated once in 100 µl 1x TdT buffer (NEB) for 30 min at room temperature, then incubated for 2 h at 37˚C in 100 µl 1x TdT buffer containing 4 µl 10 mM ATP and 1 µl Terminal Transferase (NEB M0315L). Plugs were rinsed with 1 ml Tris buffer (10 mM Tris HCl (pH 8.0)), equilibrated in 100 µl 1x T4 RNA ligase buffer (NEB) containing 40 µl 50% PEG 8000 for 1 h at room temperature, then incubated overnight at 25˚C in 100 µl 1x T4 RNA ligase buffer (NEB) containing 40 µl 50% PEG 8000, 1 µl 10 pM/µl TrAEL-seq adaptor 1 and 1 µl T4 RNA ligase 2 truncated KQ (NEB M0373L). Plugs were then rinsed with 1 ml Tris buffer, transferred to 15 ml tubes, and washed 3 times in 10 ml Tris buffer with rocking at room temperature for 1 to 2 h each, then washed again overnight under the same conditions.

DNA processing: Plugs were equilibrated for 15 min with 1 ml agarase buffer (10 mM Bis-Tris-HCl, 1 mM EDTA (pH 6.5)), then the supernatant removed and 50 µl agarase buffer added. Plugs were melted for 20 min at 65˚C, transferred for 5 min to a heating block pre-heated to 42˚C, 1 µl β-agarase (NEB M0392S) was added and mixed by flicking without allowing sample to cool, and incubation continued at 42˚C for 1 h. DNA was ethanol precipitated with 25 µl 10 M $NH_4OAc$, 1 µl GlycoBlue, 330 µl of ethanol and resuspended in 10 µl 0.1x TE. A volume of 40 µl reaction mix containing 5 µl isothermal amplification buffer (NEB), 3 µl 100 mM $MgSO_4$, 2 µl 10 mM dNTPs, and 1 µl Bst 2 WarmStart DNA polymerase (NEB M0538S) was added and sample incubated 30 min at 65˚C before precipitation with 12.5 µl 10 M $NH_4OAc$, 1 µl GlycoBlue, 160 µl ethanol and redissolving pellet in 130 µl 1x TE. The DNA was transferred to an AFA microTUBE (Covaris 520045) and fragmented in a Covaris E220 using duty factor 10, PIP 175, Cycles 200, Temp 11˚C, then transferred to a 1.5-ml tube containing 8 µl prewashed Dynabeads MyOne streptavidin C1 beads (Thermo, 65001) resuspended in 300 µl 2x TN (10 mM Tris (pH 8), 2 M NaCl) along with 170 µl water (total volume 600 µl) and incubated 30 min at room temperature on a rotating wheel. Beads were washed once with 500 µl 5 mM Tris (pH 8), 0.5 mM EDTA, 1 M NaCl, 5 min on wheel and once with 500 µl 0.1x TE, 5 min on wheel before resuspension in 25 µl 0.1x TE.

Library preparation: TrAEL-seq adaptor 2 was added using a modified NEBNext Ultra II DNA kit (NEB E7645S): 3.5 µl NEBNext Ultra II End Prep buffer, 1 µl 1 ng/µl sonicated salmon sperm DNA (this is used as a carrier), and 1.5 µl NEBNext Ultra II End Prep enzyme

were added and reaction incubated 30 min at room temperature and 30 min at 65˚C. After cooling, 1.25 μl 10 pM/μl TrAEL-seq adaptor 2, 0.5 μl NEBNext ligation enhancer, and 15 μl NEBNext Ultra II ligation mix were added and incubated 30 min at room temperature. The reaction mix was removed and discarded and beads were rinsed with 500 μl wash buffer (5 mM Tris (pH 8), 0.5 mM EDTA, 1 M NaCl), then washed twice with 1 ml wash buffer for 10 min on wheel at room temperature and once for 10 min with 1 ml 0.1x TE. Libraries were eluted from beads with 11 μl 1x TE and 1.5 μl USER enzyme (NEB) for 15 min at 37˚C, then again with 10.5 μl 1x TE and 1.5 μl USER enzyme (NEB) for 15 min at 37˚C, and the 2 eluates combined.

Library amplification: Amplification was performed with components of the NEBNext Ultra II DNA kit (NEB E7645S) and a NEBNext Multiplex Oligos set (e.g., NEB E7335S). An initial test amplification was used to determine the optimal cycle number for each library. For this, 1.25 μl library was amplified in 10 μl total volume with 0.4 μl each of the NEBNext Universal and any NEBNext Index primers with 5 μl NEBNext Ultra II Q5 PCR master mix. Cycling program: 98˚C 30 s, then 18 cycles of (98˚C 10 s, 65˚C 75 s), 65˚C 5 min. Test PCR was cleaned with 8 μl AMPure XP beads (Beckman A63881) and eluted with 2.5 μl 0.1x TE, of which 1 μl was examined on a Bioanalyser high sensitivity DNA chip (Agilent 5067–4626). Ideal cycle number should bring final library to final concentration of 1 to 3 nM, noting that the final library will be 2 to 3 cycles more concentrated than the test anyway. A volume of 21 μl of library was then amplified with 2 μl each of NEBNext Universal and chosen Index primer and 25 μl NEBNext Ultra II Q5 PCR master mix using same conditions as above for calculated cycle number. Amplified library was cleaned with 40 μl AMPure XP beads (Beckman A63881) and eluted with 26 μl 0.1x TE, then 25 μl of this was again purified with 20 μl AMPure XP beads and eluted with 11 μl 0.1x TE. Final libraries were quality controlled and quantified by Bioanalyser (Agilent 5067–4626) and KAPA qPCR (Roche KK4835).

Libraries were sequenced either on an Illumina MiSeq as 50 bp Single Read or an Illumina NextSeq 500 as High Output 75 bp Single End by the Babraham Institute Next Generation Sequencing facility.

## TrAEL-seq with barcoded adaptor for quantitative comparison

Two additional variants of TrAEL adaptor 1 were synthesised, preadenylated, and purified as for TrAEL adaptor 1 above.

Index 1: [Phos]GACTNNNNNNNNNAGATCGGAAGAGCGTCGTGTAGGGAAAGAGT GTU GCGCAGGCCATTGGCC [BtndT] GCGCUACACTCTTTCCCTACACGAC GCT [Phos]

Index 2: [Phos]AGTCNNNNNNNNNAGATCGGAAGAGCGTCGTGTAGGGAAAGAG TGTU GCGCAGGCCATTGGCC [BtndT] GCGCUACACTCTTTCCCTACACGAC GCT [Phos]

The 3′ phosphate on these adaptors was designed to prevent potential circularisation of the adaptor and is removed by the additional phosphatase treatment noted below. We do not think this modification made a substantial difference.

For preparation of libraries from G1-arrested and G1->S cells, whole agarose plugs were prepared as written above. Plugs were cut in two and each half tailed and ligated as normal, with Index 1 or Index 2 adaptor substituted for TrAEL-seq adaptor 1. This resulted in 2 ligations per sample, one with index 1 and one with index 2. Plugs were then rinsed and washed in separate 15 ml tubes, but prior to incubation with agarase buffer plugs were pooled in pairs of different conditions with opposite indexes, e.g., G1—index 1 pooled with G1->S—index 2, and vice versa. Each pool was then processed in double the volume of reagents for agarase

treatment and the first round of ethanol precipitation, followed by resuspension in 10 μL 0.1x TE. Each pooled sample was incubated with 29 μL water, 3 μL 100 mM MgSO4, 5 μL Isothermal amplification buffer, and 1 μL shrimp alkaline phosphatase (rSAP, M0371S) for 30 min at 37˚C, followed by 10 min at 65˚C. Then, 2 μL 10 mM dNTPs and 1 μL Bst 2.0 warmstart polymerase were added and incubation continued at 65˚C for 30 min. The rest of the protocol was performed as normal.

### END-seq library preparation

Note: This protocol is based on the original described by Canela and colleagues [11] but has a critical difference: The exonuclease-mediated blunting step designed for topoisomerase II ends did not work well on the 2 test substrates we use in yeast genomic DNA. Instead, best results were obtained by blunting 2 h or overnight with Klenow, which outperformed T4 DNA polymerase or a commercial DNA blunting kit.

Preparation of END-seq adaptor 1: DNA oligonucleotide was synthesised and PAGE purified by Sigma-Genosys (Merck); sequence is as described by Canela and colleagues [11]:

[Phos]GATCGGAAGAGCGTCGTGTAGGGAAAGAGTGUU[BtndT]U[BtndT]UUACA CTCTTTCCCTACACGACGCTCTTCCGATC*T

Annealed as for TrAEL-seq adaptor 2 above.

Preparation of END-seq adaptor 2c: DNA oligonucleotide was synthesised and PAGE purified by Sigma-Genosys (Merck), modified from Canela and colleagues [11] to prevent homodimers of adaptor from amplifying: [Phos]GATCGGAAGAGCTATTATTTAAATTTTAATT UGACTGGAGTTCAGACGTGTGCTCTTCCGATC*T

Annealed as for TrAEL-seq adaptor 2 above.

Sample preparation: ½ an agarose plug was used for each library (cut with a razor blade), hereafter referred to as a plug for simplicity. All incubations were performed in 2 ml round bottomed tubes (plugs break easily in 1.5 ml tubes), or 15 ml tubes for high volume washes. Restriction enzyme digestion was performed as described for TrAEL-seq.

Blunting and ligation: The plug was equilibrated for 1 h at room temperature in 100 μl NEBuffer 2 with 0.1 mM dNTPs, then blunted overnight at 37˚C in 100 μl NEBuffer 2 with 0.1 mM dNTPs and 1 μl Klenow (NEB M0210S). After rinsing twice with 1 ml Tris buffer, plug was transferred to a 15-ml tube and washed 3 times for 15 min each with 10 ml Tris buffer on rocker at room temperature before transfer to a new 2 ml tube. The plug was equilibrated with 100 μl CutSmart buffer containing 5 mM DTT and 1 mM dATP for 1 h at room temperature before incubation for 2 h at 37˚C in another 100 μl of the same buffer containing 1 μl Klenow exo- (NEB M0212S) and 1 μl T4 PNK (NEB M0201S). Plug was rinsed twice with 1 ml Tris buffer, then washed once with 10 ml of Tris buffer for 15 min as above, then returned to a 2-ml tube. The plug was equilibrated for 1 h at room temperature in 100 μl 1x Quick Ligation buffer (NEB B6058S) containing 2.7 μl END-seq adaptor 1, then overnight at 25˚C with another 100 μl of the same buffer containing 2.7 μl END-seq adaptor 1 and 1 μl high concentration T4 DNA Ligase (NEB M0202M). After rinsing twice with 1 ml Tris buffer, plug was transferred to a 15-ml tube and washed 3 times for 1 to 2 h each with 10 ml Tris buffer on rocker at room temperature, then again overnight.

DNA purification and library construction: The plug was transferred to a 1.5-ml tube and equilibrated 15 min with 1 ml agarase buffer (10 mM Bis-Tris-HCl, 1 mM EDTA (pH 6.5)), then the supernatant removed and 50 μl agarase buffer added to the plug. Plug was melted 20 min at 65˚C, then transferred for 5 min to a heating block preheated to 42˚C, 1 μl beta-agarase (NEB M0392S) was added and mixed by flicking without allowing sample to cool, and incubation continued at 42˚C for 1 h. DNA was ethanol precipitated with 25 μl 10 M NH4OAc, 1 μl

GlycoBlue, 330 μl of ethanol and resuspended in 130 μl 1x TE, 15 min at 65˚C. From here, samples were sonicated, purified, and library construction performed as for TrAEL-seq, except that END-seq adaptor 2c was substituted for TrAEL-seq adaptor 2.

## In vitro TrAEL activity and qPCR assays

For in vitro assays, 0.5 μl 10 μM DNA oligonucleotide CGCGGTAATTCCAGCTCCAA was treated with or without 0.5 μl TdT in 20 μl 1x TdT buffer containing 0.8 μl 10 mM ATP for 30 min at 37˚C. Reactions were purified by phenol:chloroform extraction and ethanol precipitation and resuspended in 5 μl 10 mM Tris (pH 8). This was ligated to 1 μl TrAEL-seq adaptor 1 in 20 μl 1x T4 RNA ligase buffer containing 8 μl 50% PEG 8000 and 1 μl T4 RNA ligase 2 truncated KQ overnight at 25˚C. Reactions were resolved on a 15% PAGE/8 M urea gel and stained with SYBR Gold (Thermo S11494) as per manufacturer's instructions.

## Data analysis

Unique Molecular Identifier (UMI) deduplication and mapping: Scripts used for UMI handling as well as more detailed information on the processing are available here: https://github.com/FelixKrueger/TrAEL-seq). Briefly, TrAEL-seq reads are supposed to carry an 8-bp in-line barcode (UMI) at the 5′-end, followed by a variable number of 1 to 3 thymines (T). Read structure is therefore NNNNNNNN(T)nSEQUENCESPECIFIC, where NNNNNNNN is the UMI, and(T)n is the poly(T). The script TrAELseq_preprocessing.py removes the first 8 bp (UMI) of a read and adds the UMI sequence to the end of the readID. After this, up to 3 T (inclusive) at the start of the sequence are removed. Following this UMI and Poly-T preprocessing, reads underwent adapter and quality trimming using Trim Galore (v0.6.5; default parameters; https://github.com/FelixKrueger/TrimGalore). UMI-preprocessed and adapter-/quality-trimmed files were then aligned to the respective genome using Bowtie2 (v2.4.1; option:—local; http://bowtie-bio.sourceforge.net/bowtie2/index.shtml) using local alignments. Finally, alignment results files were deduplicated using UmiBam (v0.2.0; https://github.com/FelixKrueger/Umi-Grinder). This procedure deduplicates alignments based on the mapping position, read orientation, as well as the UMI sequence.

For samples carrying sample-level barcodes, the read structure is NNNNNNNNBBBB(T)nSEQUENCESPECIFIC, where NNNNNNNN is the UMI, BBBB is the sample barcode (currently either AGTC or GACT), and(T)n is the poly(T). A script handling the preprocessing of these libraries is available from the code repository (https://github.com/FelixKrueger/TrAEL-seq/blob/master/TrAELseq_preprocessing_UMIplusBarcode.py).

UMI deduplicated mapped reads were imported into SeqMonk v1.47 (https://www.bioinformatics.babraham.ac.uk/projects/seqmonk/) and immediately truncated to 1 nucleotide at the 5′ end, representing the last nucleotide 5′ of the strand break. Reads were then summed in running windows or around features as described in figure legends. Windows overlapping with non-single-copy regions of the genome were filtered (rDNA, 2μ, mtDNA, *CUP1*, subtelomeric regions, Ty elements and LTRs), and total read counts across all included windows were normalised to be equal. Scatter plots and average profile plots were generated in SeqMonk, and in the latter case, the data were exported and plots redrawn in GraphPad Prism 8.

For read count quantification and read polarity plots, data were first imported into SeqMonk v1.47 and truncated to 1 nucleotide as described above. Reads (total or separate forward and reverse read counts) were quantitated in running windows as specified in the relevant figure legends before export for plotting using R v4.0.0 in RStudio using the *tidyverse* package [89,90]. For displaying read counts, values were plotted at the centre of the quantification

window and displayed as a continuous line. For read polarity plots, read polarity values were calculated and plotted as either dots (individual samples) or as a continuous line (multiple sample display) for each quantification window using the formula read polarity = $(R - F)/(R + F)$, where F and R relate to the total forward and reverse read counts respectively. The R code to generate these plots can also be found here: https://github.com/FelixKrueger/TrAEL-seq.

A note on read polarity: As a consequence of experimental design, the Illumina sequencing read is the reverse complement of the 3′ extended DNA to which TrAEL adaptor 1 was ligated, and so the first nucleotide of the read is the reverse complement of the last nucleotide 5′ of the break site. To minimise potentially confusing strand inversions, we did not invert the reads during the analysis. In contrast, Sriramachandran and colleagues reversed the polarity of all reads in the analysis pipeline for GLOE-seq [40], which explains the differences in polarity between equivalent analyses in that study and this study. The relationships between the libraries and read mapping statistics are summarised in S2 Table.

## Supporting information

**S1 Fig. TrAEL-seq library construction details. (A)** Example Bioanalyzer trace for the amplified library of *Not*I *Pme*I *Sfi*I-digested yeast genomic DNA. A volume of 1 μl of the 10.5 μl final library was run on a DNA high sensitivity Bioanalyzer chip. This shows a complete absence of adaptor or primer dimers, which is only achieved after 2 successive AMPure purifications. This trace is typical for TrAEL-seq libraries. (**B**) Schematic of TrAEL-seq read processing pathway. TrAEL-seq reads are the reverse complement of the original DNA end. The 8 nucleotide UMI is removed and stored, then up to 3 T's are removed from the 5′ of the read. Poor-quality reads and adaptor sequences are removed by TrimGalore, then reads are mapped using Bowtie 2. Deduplication is performed based on the UMI and the mapped start site by UMI grinder, then the reads are finally truncated to a single nucleotide representing the reverse complement of the terminal nucleotide of the original DNA strand. (**C**) Quantitation of DNA ends generated by *Sfi*I digestion categorised by the 3′ nucleotide or the nucleotide adjacent to the 3′ nucleotide in TrAEL-seq data. Bars show mean and 1 SD. (**D**) Precision mapping of *Sfi*I cleavage sites by TrAEL-seq and END-seq, as Fig 1D. This graph represents the 10 *Sfi*I sites that have 2 or more As at the 3′ end (GGCCNNAA|NGGCC). In this category are 5 ends with 2 As, 2 ends with 3 As, and 3 ends with 4 As. Mapped locations of 3′ ends were averaged across each category of site and expressed as a percentage of all 3′ ends mapped by each method to that category of site. (**E**) Scatter plot of log-transformed normalised read counts at all 3,907 Spo11 cleavage hotspots annotated by Mohibullah and Keeney [1], comparing 2 technical replicate TrAEL-seq libraries generated from the same sample of *dmc1Δ* cells. The 2 libraries were prepared approximately 6 months apart by 2 different researchers from cells stored in 70% ethanol at −70˚. (**F**) Scatter plot of log-transformed normalised read counts at all 3,907 Spo11 cleavage hotspots annotated by Mohibullah and Keeney, comparing *dmc1Δ* TrAEL-seq with data for Spo11-associated oligonucleotides [1–4] (SRA accession: SRR1976210). Numerical data underlying this figure can be found in S1 Data. TrAEL-seq, Transferase-Activated End Ligation sequencing; UMI, unique molecular identifier.
(TIF)

**S2 Fig. Additional data for detection of replication fork stalling by TrAEL-seq. (A)** Reproducibility of RFB detection between 2 technical replicates. The 2 libraries were prepared approximately 6 months apart by 2 different researchers from cells stored in 70% ethanol at −70˚. (**B**) Detection of RFB peaks without nonreproducible background peaks in 3 biological replicates TrAEL-seq libraries derived from wild-type cells. (**C**) Replication direction of centromeres, calculated based on the *cdc9*-AID GLOE-seq data (SRA accession: SRX6436838).

Percentage of reverse reads was determined in the regions −1000 to −500 bp and +500 to +1000 bp relative to the annotated centromere, and the average of these values plotted. The region from −500 to +500 bp was excluded as replication fork stalling in this region obscures the replication direction. CEN2 is misleading as it is directly adjacent to a replication origin— see S1 File for profiles of individual centromeres. (**D**) Average TrAEL-seq profiles across tRNAs ±200 bp for 2 biological replicates of hESC cells, each averaged from 2 technical replicates. Reads are separated by orientation on forward or reverse strands; all tRNAs are included. Read counts per million reads mapped were calculated in nonoverlapping 5 bp bins. (**E**) Average TrAEL-seq profiles across all centromeres ±1 kb for wild-type and *rad52Δ* cells. Read counts per million reads mapped were calculated in nonoverlapping 10 bp bins. (**F**) Average TrAEL-seq profiles across all tRNAs ±200 bp for wild-type and *rad52Δ* cells. Read counts per million reads mapped were calculated in nonoverlapping 5 bp bins. Numerical data underlying this figure can be found in S2 Data. hESC, human embryonic stem cell; RFB, replication fork barrier; TrAEL-seq, Transferase-Activated End Ligation sequencing. (TIF)

**S3 Fig. Additional data for replication fork directionality of TrAEL-seq data.** (**A**) Scatter plot showing the percentage of reverse reads compared to all reads in 1 kb genomic windows spaced every 1 kb, comparing TrAEL-seq data from wild-type cells and GLOE-seq data from Cdc9-depleted cells (SRA accession: SRX6436838). (**B**) Read polarity plots showing TrAEL-seq data for wild type, *fob1Δ*, and *rad52Δ* across a single rDNA repeat. The 35S rRNA gene transcribed by RNA polymerase I is shown as a thicker grey line and is transcribed right to left in this representation. Mature rRNA genes are shown in black; the RFB and the ARS are also annotated. Inset is the region containing the RFB sites that is shown in Fig 2B. (**C**) Scatter plot showing the percentage of reverse reads compared to all reads in 1 kb genomic windows spaced every 1 kb, comparing TrAEL-seq data from 2 technical replicates of wild-type cells. (**D**) Read polarity plot across chromosome V for TrAEL-seq datasets of wild type compared to the RNase H2 mutants *rnh201Δ* and *rnh202Δ* and topoisomerase I mutant *top1Δ*. (**E**) Read polarity plot for chromosome V comparing END-seq and TrAEL-seq data generated from two-halves of an agarose plug containing 10 million wild-type 3x*CUP1* cells grown in synthetic complete glucose media. Note that the scale for the END-seq data is expanded as the bias in read polarity is much smaller in END-seq libraries. (**F**) Scatter plot showing the percentage of reverse reads compared to all reads in 250 kb genomic windows spaced every 10 kb, comparing TrAEL-seq data for 2 technical replicates generated from the same hESC sample. (**G**) Scatter plot showing the percentage of reverse reads compared to all reads in 250 kb genomic windows spaced every 10 kb, comparing TrAEL-seq data for 2 biological replicates of hESCs, each averaged from 2 technical replicates. (**H**) Scatter plot showing the percentage of reverse reads compared to all reads in 250 kb genomic windows spaced every 10 kb, comparing TrAEL-seq data from hESC cells (average of 2 technical replicates) to GLOE-seq data from LIG1-depleted HCT116 cells (average of SRA accessions: SRX7704535 and SRX7704534). Numerical data underlying this figure can be found in S3–S6 Data. ARS, autonomously replicating sequence; hESC, human embryonic stem cell; RFB, replication fork barrier; TrAEL-seq, Transferase-Activated End Ligation sequencing. (TIF)

**S4 Fig. Additional data for detection of environment-dependent replication differences.** (**A**) Scatter plot showing the percentage of reverse reads compared to all reads in 1 kb genomic windows spaced every 1 kb, comparing TrAEL-seq data wild type and *clb5Δ* (left). An equivalent comparison between wild type and *rnh201Δ* (which has a wild-type replication profile) is shown for comparison (right). (**B**) Plot of read count across the GAL locus on galactose

induction for *dnl4Δ rad51Δ* mutant, as Fig 4B. (**C**) MA plots of changing read count across the genome on galactose induction for *dnl4Δ rad51Δ* mutant, as Fig 4C. (**D**) Read polarity plots showing the replication profile of the region surrounding the GAL locus with and without galactose induction. Green box shows the site at which the replication fork which passes through the GAL locus encounters the oncoming fork from *ARS211*. (**E**) Plot of average TrAEL-seq read density around the TSS in the highest 25% expressed genes orientated head-on with replication (as Fig 4D). Data are shown for G1 and G1->S samples (Fig 3E); genes are averaged together within each sample, but the difference in average read count between samples is maintained. The nonreplicating G1 sample contains far less reads on average across TSS regions, and the peak upstream of the TSS is absent. Numerical data underlying this figure can be found in S7 Data. TrAEL-seq, Transferase-Activated End Ligation sequencing; TSS, transcriptional start site.
(TIF)

**S5 Fig. Means by which reversed forks could resemble DSBs in southern analysis.** All Southern blot analyses that have reported direct detection of DSBs at RFBs utilise a restriction digestion to separate the region of interest. For the yeast RFB, to our knowledge, the enzyme used has always been *Bgl*II, the cleavage sites for which lie 2.2 kb and 2.4 kb each side of the RFB. Forks that reverse past the *Bgl*II site would yield a *Bgl*II fragment the same size (2.2 kb) as a fork that is cleaved at the RFB. Only fragments that would hybridise to the probe (blue) are shown. DSB, double-strand break; RFB, replication fork barrier.
(TIF)

**S1 Table. Yeast strains used in this study.**
(XLSX)

**S2 Table. List of all libraries produced during this work, including GEO accession and mapping statistics.**
(XLSX)

**S1 File. TrAEL-seq profiles at individual centromeres.**
(PDF)

**S2 File. Detailed TrAEL-seq protocol.**
(DOC)

**S1 Data. Underlying numerical data.**
(XLSB)

**S2 Data. Underlying numerical data.**
(XLSB)

**S3 Data. Underlying numerical data.**
(XLSB)

**S4 Data. Underlying numerical data.**
(XLSB)

**S5 Data. Underlying numerical data.**
(XLSB)

**S6 Data. Underlying numerical data.**
(XLSB)

**S7 Data. Underlying numerical data.**
(XLSB)

**S1 Raw Images. Raw gel image.** Note that not all lanes are presented in the manuscript.
(PDF)

## Acknowledgments

We thank Paula Koko Gonzales and Nicole Forrester of the Babraham Institute Next Generation Sequencing facility for data generation, Scott Keeney for sharing unpublished data, Adele Marston and Aziz El Hage for yeast strains, Stephen Bevan for growing cells, and New England Biolabs technical support for helpful answers to a wide range of enzymology questions during the development of this method.

## Author Contributions

**Conceptualization:** Jonathan Houseley.

**Funding acquisition:** Jonathan Houseley.

**Investigation:** Neesha Kara, Peter Rugg-Gunn, Jonathan Houseley.

**Methodology:** Jonathan Houseley.

**Software:** Neesha Kara, Felix Krueger.

**Supervision:** Jonathan Houseley.

**Visualization:** Neesha Kara.

**Writing – original draft:** Jonathan Houseley.

**Writing – review & editing:** Neesha Kara, Felix Krueger, Peter Rugg-Gunn.

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
