## [Editor Report · Decision Letter 0]

4 Aug 2020

Dear Jon, 

Thank you for submitting your manuscript entitled "Genome-wide analysis of DNA replication and DNA double strand breaks by TrAEL-seq" for consideration as a Methods and Resources by PLOS Biology.

Your manuscript has now been evaluated by the PLOS Biology editorial staff, as well as by an academic editor with relevant expertise, and I'm writing to let you know that we would like to send your submission out for external peer review.

Please re-submit your manuscript within two working days, i.e. by Aug 06 2020 11:59PM.

Kind regards,

Roli

Senior Editor

PLOS Biology

---

## [Decision Letter · Decision Letter 1]

21 Sep 2020

Dear Jon,

Thank you very much for submitting your manuscript "Genome-wide analysis of DNA replication and DNA double strand breaks by TrAEL-seq" for consideration as a Methods and Resources paper at PLOS Biology. Your manuscript has been evaluated by the PLOS Biology editors, an Academic Editor with relevant expertise, and by three independent reviewers.

You’ll see that all three reviewers are broadly positive about the study, but raise a number of overlapping concerns that will need to be addressed, some involving additional experimental work (e.g. both revs #1 and #3 suggest running TrAEL-seq after alpha-factor arrest); there are also a range of textual and presentational requests.

In light of the reviews (below), we will not be able to accept the current version of the manuscript, but we would welcome re-submission of a much-revised version that takes into account the reviewers' comments. We cannot make any decision about publication until we have seen the revised manuscript and your response to the reviewers' comments. Your revised manuscript is also likely to be sent for further evaluation by the reviewers.

We expect to receive your revised manuscript within 3 months. 

**IMPORTANT - SUBMITTING YOUR REVISION**

*Re-submission Checklist*

*Published Peer Review*

*PLOS Data Policy*

*Blot and Gel Data Policy*

Best wishes,

Roli

Senior Editor,

rroberts@plos.org,

PLOS Biology

REVIEWERS' COMMENTS:

Reviewer #1:

The authors introduce a method for the genome-wide detection of DNA 3' ends that they call TrAEL-seq. The initial experiments clearly demonstrate the sensitivity of the approach to detect free 3' ends generated by restriction enzyme cuts. The authors then go on to assess the pattern of meiotic double strand breaks, sites of replication fork stalling and the pattern of DNA replication. Intriguingly, and somewhat frustratingly, the authors don't have an explanation for why TrAEL-seq can be used to determine the direction of replication forks. Overall, TrAEL-seq looks to be a useful addition to the wide range of methods available to assess DNA ends and DNA replication.

I have a number of comments that I hope might help the authors improve their manuscript:

1. This manuscript is primarily a methods paper and therefore I think it is important that a number of additional controls are considered:

- for the second half of the manuscript the authors use TrAEL-seq to assess DNA replication. However, I didn't spot an experiment that shows that the signal the authors are detecting is DNA replication dependent. A simple TrAEL-seq on arrested yeast cells (e.g. in alpha-factor) would provide clear evidence that the output reported is replication dependent. There are multiple places in the manuscript where the authors describe the signal and state that it is DNA replication dependent, however without a non-replicating control I don't see how they can support these statements. Furthermore, there is likely to be some level of background (there is to any genomics method) and a non-replicating control should allow the authors to assess this. This would be particularly important when the authors compare mutants or growth conditions (glucose vs. other carbon sources), since the fraction of S phase cells is very likely to be different and therefore background signal could be influencing the results.

- that authors use restriction with SfiI to introduce the method. These were well thought through and well executed experiments. This is an enzyme that leaves a 3' overhang. At various points through the manuscript the authors mention that TrAEL-seq has differing sensitivities to various 3' ends (e.g. figure 2A coloured dots). It would be useful for the authors to properly assess this by performing a TrAEL-seq experiment with a cocktail of various enzymes that leave different produces (nicking, 5' overhand, 3' overhand, blunt ends).

- the authors undertake a comprehensive analysis of the bases detected at the 3' end of the SfiI digested material. I would like to see the authors undertake a comparable analysis to see if there is any preference for the base at or close by to the detected 3' end when TrAEL-seq was applied to various genomic DNA substrates. 

- how much does the TrAEL-seq read density across the genome vary? In Fig. 3B (and similar figures) the authors present the strand bias, but not the total number of reads mapping to each region. This would be a useful analysis, particularly when considering what might be the source of the 3' ends detected by TrAEL-seq. In figure 4B the density on the reverse strand is shown and appears to have some periodicity; I'd encourage the authors to look further as to the nature and potential source of this periodicity. 

2. The authors don't discuss the difference between wild-type and rad52-delta cells at centromeres (rad52-delta gives reduced peaks) and tRNA genes (rad52-delta seems identical to wild-type). Why is the signal at centromeres reduced in the rad52 mutant?

3. The authors speculate as to why TrAEL-seq has the ability to report on replication fork direction, but (as they clearly state) they don't have a confirmed molecular mechanism. As such, I think they should be more careful in their terminology in some places, particularly in referring to the y-axis of plots such as Fig 3B, D, F & G as "Replication Fork Direction". They are not measuring replication fork direction (and don't have a explanation for why what they are measuring correlates with replication fork direction) and therefore this shouldn't be used to label these axis. The authors should find an alternative description, such as "strand bias" for the manuscript text and figure labels.

4. Are the authors confident that RNase T1 (used to treat DNA in plugs) cannot digest adjacent to ribose bases incorporated into genomic DNA? If it can this could offer an alternative explanation for the observed strand bias that allows the authors to infer replication fork direction. However, it would also require the authors to reassess the rnh201-null/rnh202-null experiments, since in this case they would not be reporting what the authors think they are.

5. What is the difference between the dnl4/rad51 double mutant and rad52-delta data?

6. In the results the authors suggest that "the replication fork is stalled by chromatin or proteins bound at the promoter" when the GAL genes are induced. The authors should test this hypothesis by looking in their other datasets at the promoters of constitutively highly expressed genes.

7. The authors have a highly plausible explanation for how 3' ends might report replication fork direction: "The mechanism responsible must frequently rearrange replication forks to make the leading strand 3' end accessible to TdT while the lagging strand 3' end remains largely inaccessible. We suggest that transient fork reversal would have this effect..." First, the use of the word "must" here seems too strong given how many unknowns remain (for example, see suggested control experiments above). Second, I suspect that this transient fork reversal is more likely to take place in the plug after Proteinase K treatment rather than in cells. The DNA is treated with Proteinase K at 50 °C which should allow for transient melting of the free 3' end (on the leading strand) and thus interconversion between the species drawn out in figure 5. Both the elevated temperature and the removal of proteins would seem to make this interconversion more likely than in cells.

Finally, the authors use green and red dots to distinguish certain features (fig 2A & 5), plus possibly green and orange violins (fig 4E). I'm red/green colour-blind and the colours used are indistinguishable to me. I don't know what the PLoS family of journals policy is on the use of colours accessible to all, but I'd encourage the authors to change this. More information and advise is available here:

https://thenode.biologists.com/data-visualization-with-flying-colors/research/

Minor comments:

- introduction: "This can be a problem as end resection forms extended tracts of 3' extended single stranded DNA..." I think this could be more clearly phrased to avoid using the word "extended" twice.

- introduction: "However, these methods do not have the resolution to detect individual origins unless markedly different in timing, and a range of other more specialised approaches have been applied to study replication initiation..." This is not correct. In budding yeast (the organism used for most of the presented work) copy number approaches are more than sufficient to detect individual replication origins. Furthermore, copy number approaches are very much more straightforward than the approach presented in this paper.

- results: "This was ligated with ~10% efficiency to pre-adenylated TrAEL-seq adaptor 1 using truncated T4 RNA ligase 2 KQ, a ligase that is specific for 5' adenylated adaptors (Fig. 1B)". I did not understand what the authors were doing here; why was the adaptor pre-adenylated and why are they interested in a 5' adenylated adaptor when TrAEL-seq uses a 3' adenylated substrate?

- results: "...consistent with previous studies that reported both co-directional and head-on tRNA transcription can stall replication forks, at least in the absence of replicative helicases (Fig. 2E arrows) [65-67]." The authors have missed a pre-genomics study from Carol Newlon's group:

Deshpande, A. M., & Newlon, C. S. (1996). DNA replication fork pause sites dependent on transcription. Science (New York, NY), 272(5264), 1030-1033. http://doi.org/10.1126/science.272.5264.1030

This paper uses DNA 2D gels to detect replication fork pausing at a tRNA genes in wild-type cells.

- results: "There should never be less 3' ends on the lagging strand..." less should be fewer.

- results: "ARS211" should be in italics.

- discussion "existing methods profile either fork direction or origin timing" - I think that recently developed single molecule nanopore methods, such as D-NAscent, can profile both fork direction and origin timing.

- fig 1A: it would be useful for the authors to label at least a couple of the free ends with 5' and 3'.

- fig 1B: which lanes contain substrate? What's the difference between lanes 4 & 5? Or 0 and 2?

- fig 1E: I think that a further 'zoom in' on one of the peaks would be a valuable addition.

- fig 2D: why have the authors presented a meta-analysis of the signal across the various classes of centromeres? It would be useful for the authors to show each individual centromere in the supplement.

- fig 3G: why have the authors reversed the y-axis for the human GLOE-seq data, but not the yeast?

- fig 4B: it would be valuable for the authors to show the forward strand reads in addition to the reverse strand reads.

- fig 4C: I'm not sure I fully understand how the data has been normalised for these two plots. What is responsible for the asymmetry? There seem to be many more locations where the read count is lower (e.g. below -50) than higher (e.g. above +50).

- fig S1A: a scale bar should be added.

- fig S1B: it is not clear what the x-axis scale is. Is this the difference in cycle number compared to the control?

- fig S2B: why is there such a large difference in peak height between the two conditions?

Reviewer #2:

In this manuscript, Kara and colleagues describe a novel NGS-based assay to map DNA double-strand breaks called Transferase-Activated End Ligation sequencing (TrAEL-seq). This method adds to the long list of assays (BLESS, BLISS, Break-seq, END-seq, i-BLESS, GLOE-seq…) that have been recently developed to map single- and double-strand DNA breaks and have revolutionized the fields of DNA repair and genome stability. The authors convincingly show that this method can efficiently capture single stranded DNA 3' ends genome-wide in yeast and in human cells. From this respect, this method differs from BLESS and related assays (BLISS, END-seq, i-BLESS…) in that it maps the real cleavage site and does not require the degradation of the 3' overhang, which can be several kb long. Yet, TrAEL-seq is not just another method to map DSBs. Indeed, the authors show that TrAEL-seq provides also a very sensitive and accurate method to map replication fork direction (RFD) in unsynchronized populations of cells. The quality of RFD maps provided for yeast and human ES cells is very impressive. It seems to match the performance of more labor-intensive assays such as OK-seq without requiring the use of ligase mutants. Moreover, the authors show that TrAEL-seq can be used to map programmed pause sites at the rDNA of yeast and human cells and at tRNA genes with an unprecedented resolution. They propose that the ability of TrAEL-seq to map paused and moving forks is due to frequent fork reversal, exposing a 3' overhang on the leading strand that can be extended by TdT to capture it. 

Beyond the validation of the assay, the manuscript provides convincing evidence that transcription does not induce replication fork pausing in budding yeast, which remains a controversial issue (see specific issues below). It also shows that changes in carbon sources in the growth medium alters the position of replication termination sites, presumably by altering the timing of origin firing. It also shows that the replication fork barrier is polar in hESCs, as it is the case in yeast and unlike in human cancer cell lines. 

Overall, the manuscript is very well written and the data are clear and convincing. The experimental pipeline is well described and the detailed protocol provided in supplementary materials will be very useful to those interested in implementing TrAEL-seq in their laboratories. Importantly, the authors also provide advices on how to combine TrAEL-seq with other methods and stress the fact that many of the existing DSB mapping assays are complementary and should be combined to embrace the complexity of DNA damage and repair processes. In conclusion, I have no doubt that this novel assay will be quickly adopted by a wide community and especially to those working on DNA replication and genomic instability. However, the following issues need to be addressed before publication. 

Specific issues:

1. Page 1 and 2, the authors review the existing NGS-based assays to map DNA breaks but omitted Break-seq, an assay developed by the Feng lab (Hoffman et al, 2015, Genome Research 25, 402). Is there any reason for that? If not, it would be fair to mention this method together with the others.

2. Modifications of BLESS (page 2) have not only improved ligation efficiency, but also increased the signal-noise ratio. As discussed in ref 15, the main limitation of the original BLESS assay was a very high noise caused by artifacts associated with formaldehyde fixation. 

3. One of the most striking features of TrAEL-seq is its sensitivity for replication forks. The author's explanation that limited fork reversal occurs very frequently even at unchallenged forks is plausible. Another likely possibility that is not discussed by the authors is that branch migration or partial unwinding of nascent DNA occurs after DNA extraction. Would it be possible to experimentally address this possibility? 

4. Figure 4B shows that highly expressed PolII genes do not induce replication fork pausing, unlike tRNA and rRNA genes. This observation argues against earlier studies in budding yeast (e.g. Azvolinsky et al., 2009, Mol. Cell 34, 722) but is consistent with a more recent study showing that spontaneous replication stress in yeast is not caused by replication-transcription conflicts (Forey et al., 2020, Mol Cell 78, 396). This issue should be discussed in the manuscript. 

5. Figure 4B also shows that breaks occur at the promoter of GAL7 and GAL10 genes upon galactose induction. Do these breaks depend on DNA replication or are they also detected in G1 cells? 

6. Although the conclusion that ~2% of the yeast genome shows altered replication direction between glucose and non-glucose media is valid, the way Fig. 4E was built is unclear. This figure aims at comparing RFD in cells grown on different carbon sources "by filtering windows for those with a difference in RFD>0.4 between the two sets". However, several strains grown on glucose (fob1, rad52, rnh201, rnh202, clb5) were not tested on raffinose and reciprocally for dln4 rad51, so it is not clear which two sets were compared to select windows with a RFD>0.4. For instance, what is clb5 grown on glucose compared to? If it is to wild type cells on raffinose, it is misleading to assume that the difference observed is due to the carbon source. Since samples are not labeled in the figure, it gives the impression that authors are comparing apples and pears. The authors should rather restrict the main figure to wild type cells grown on glucose, raffinose and raffinose + galactose and move the figure with the whole set of data to supplementary material after clearly explaining how samples were compared to each other. Focusing the main figure on wild type cells would also stress the fact that the main determinant of this RFD difference is the absence of glucose and not the induction of GAL genes. 

Reviewer #3:

General comments:

1. The overall clarity of the paper could be improved by making it clear in Figure 1A that TrAEL-seq reads are the opposite polarity of the actual inferred 3' end. Throughout the paper, I was occasionally confused by this, until noticing in figure caption 1A that "TrAEL-seq reads map antisense to the cleaved strand, reading the complementary sequence starting from the first nucleotide before the cleavage site.". This description itself is confusing and would be clarified with a better and more complete figure in Figure 1A (and elsewhere.

This confusion arises again later when comparing to GLOE-seq data. 

2. Abstract. Please revise extensively to make the text appropriate for a general non-specialist audience (e.g. PLoS Biology). The opening the abstract requires some general information about DNA replication (what it is, what it is for, what the problems/questions are) that sets the scene. As written it jumps in too fast to a technical aspect of DNA replication fork stalling. Indeed, to a non-specialist it is not even clear the text is talking about *DNA* replication.

3. In general, please take time to split the text up more into smaller paragraphs with more headings, and make sure each experimental section is appropriately introduced. As currently written, it will confuse a lot of the PLoS Biology readership. Currently, too many things are left unstated. Please be explicit.

---

Page 8. Major comments:

1. The authors contend that they are able to map nascent replication forks. If this is correct, the authors should test by preparing libraries in alpha-factor arrest (or G2/M arrest). A more complex experimental test (that may not work) would be a release into S-phase and demonstrate both that signals arise and also move in the expected direction of replication.

2. The authors suggest that they are detecting transient instances of fork reversal. Whilst possible, it strikes me as highly unlikely that such events would be so frequent and so prevalent as to generate the data in Fig 3. I would like to suggest an alternative explanation: That they are detecting instances of fork collision with Top1 on the leading strand ahead of the fork. Please refer to: https://www.ncbi.nlm.nih.gov/pmc/articles/PMC85758/

The above paper demonstrates how only Top1 CCs on the leading strand template generate DSBs. Indeed this model could explain the strand disparity observed by TrAEL-seq. (Note that the Top1-CC would prevent detection of signal with the opposing polarity, because the Top1-CC would prevent ligation.).

3. The authors do not appear to have normalised their data for relative sequencing depth; thus any comparisons between libraries/methods are circumspect and misleading as presented. Please: A) Tabulate the read depth and mapping/filtering etc of all libraries. B) Present the consistency/correlation of repeat TrAEL-seq datasets to justify pooling. (I assume there are biological duplicates and that they were pooled in some way?)

---

Other specific comments:

Page 2. "DNA breaks cannot therefore be mapped post-resection by BLESS-type methods, which is problematic as DSB repair is often easiest to inhibit post-resection (such as in classic rad51Δ or rad52Δ mutants in yeast)."

This statement needs revising: This is absolutely not true (as written). DSBs can be (and have been!) mapped, but lack nucleotide precision of the original site.

Page 2. "Profiles yielded by BLESS-type methods can rarely be considered in isolation as replication has a dramatic influence on the distribution of DNA strand breaks in a cell;"

Is there a reference to back up this very strong ("dramatic influence") statement? Has BLESS (or other) been performed in strains arrested versus going through replication? If so, please reference, and or amend the statement to make it clear that it is the authors' view that the distribution will be altered based on knowledge of fork-stalling/blocks etc.

Page 2. "Methods have also been developed to detect replication fork directionality through isolation and sequencing of Okazaki fragments (OK-seq) [32, 33]"

Since a fairly exhaustive discussion of general methods are being presented, I suggest also including Pu-Seq (https://www.ncbi.nlm.nih.gov/pmc/articles/PMC4789492).

Page 2. Please hyphenate "single-stranded DNA".

Page 3. First two paragraphs. The structure of the paper here is very confusing. It opens with a description/overview of the method (and mentions the use of T4 RNA ligase), but then in paragraph 2 'implementation", the text reads as if the RNA ligase has not yet been mentioned. This is very confusing. Please revise the text so that the order is logical and suitable for a broad readership. In general, many of the paragraphs/sentences could be greatly improved with simple leaders such as: " In order to do X, we did Y...".

Page 3. Is the efficiency of tailing and ligating to a (very) short 18 nt oligo a suitable way to measure the efficiency of tailing and ligating to rare ssDNA ends within large genomic DNA fragments? What is the estimated relative molarity of ends in the test vs real reaction?

MINOR: Fig 1 legend. Please don't refer to a ssDNA as 18 bp (base pairs) in length when there is no base pairing. It is 18 nucleotides (nt) in length.

Page 3. Please define ∆Ct in the main text. To maximise accessibility, please avoid all unnecessary jargon and abbreviations.

Page 3 "DNA was digested prior to ligation with the restriction enzyme SfiI, then ΔCt was calculated for qPCR reactions that detect TrAEL adaptor 1 ligated to an SfiI cleavage site in the ribosomal DNA (rDNA) (Fig. S1A)."

Confusing sentence structure, please revise to: 

"DNA was digested with the restriction enzyme SfiI prior to ligation TrAEL adaptor 1, then ΔCt was calculated for qPCR reactions that detect TrAEL adaptor 1 ligated to an SfiI cleavage site in the ribosomal DNA (rDNA) (Fig. S1A)."

Page 3. "Data from two experiments was compared with two libraries previously

generated using an END-seq protocol..."

This is very unclear. What is meant by previously generated? Are these published data from somewhere? Or new by the lab? If this is the first time these data have been presented, why does the text state "Previously generated"? What point is being made by such a statement?

Minor. Figure 1B, there is no description of what the short vertical bars and dots indicate. (I presume means and raw datapoints.)

Page 3. The final paragraph is very technical and hard to follow. The figure legend for S1C does not help at all. In addition, Presentation of a single bioanalyser trace in Fig S1D is unhelpful if the point is to demonstrate primer removal: the input lanes need to also be shown. (Note, I'm really not sure that this panel and the text is needed at all...but in its current form it is of very limited value.)

Was the genomic DNA digested to completion? Or was it partial? Where is the raw data supporting this?

Page 4. "Comparing TrAEL-seq data for SfiI digested genomic DNA to an END-seq library generated from equivalently digested material shows high concordance (Fig. 1C)."

To aid reader comprehension, the opening sentence of this section should first indicate that TrAEL-seq libraries were sequenced, then briefly describe how the data were processed (i.e. mapping, genome, tools used, and any filtering etc) before comparing with END-seq data. This is especially confusing (as currently written) since all prior data presentation concerned qPCR at specific test loci.

Page 4. What does "unambiguously" refer to here? How would the data look if the 70 SfiI sites had been detected "ambiguously"?

Page 4. Without further details about how mapping, trimming and/or filtering were performed (summarised clearly in the main text), interpretation of the nucleotide accuracy is circumspect. For example, How many rA bases are assumed to have been added? Are all T bases trimmed prior to (or after) mapping?

Page 4. "We suggest that this overall mapping accuracy of >99% within ±1 nucleotide would be sufficient for almost all applications."

This estimation is only relevant for DSBs generated by SfiI. DSBs from other sources may form at loci with longer runs of genome-encoded adenosines, where the algorithm may not perform as well. (Actually, what is the algorithm that has been used? Why is it not explained in the main text?) This sentence is therefore misleading, as it implies that the algorithm accuracy (">99% within ±1 nucleotide") is true for "almost all applications". Please address this by estimating accuracy at sites with longer runs of genomic-encoded adenosines or by revising the interpretation/presentation of this accuracy estimation (and by explaining how this algorithm works).

Page 4. "A major strength of TrAEL-seq should be the ability to map DSB sites after resection," AND "This shows that TrAEL-seq accurately maps endogenous resected DSBs."

These sentences could be clarified to indicate that TrAEL-seq maps the original site of DSB formation, rather than the endpoint of resection. 

Page 4. "Meiotic DSBs formed by Spo11 are processed by Sae2 amongst other factors prior to resection, after which strand invasion into a sister chromatid is mediated by Dmc1 [40, 41]."

In meiosis, homologous recombination occurs between homologous chromosomes, rather than sister chromatids. Please revise.

Page 4. "TrAEL-seq for resected DSBs in dmc1Δcells 7 hours after induction of meiosis revealed a DSB pattern very similar to that observed forunresected DSBs in an sae2Δ mutant mapped by S1-seq (a BLESS variant specific for meioticrecombination) (Fig. 1E)."

Please cite the paper from which the sae2D S1-seq data originate.

Page 4 and Figure 1F. "Across all hotspots for Spo11 cleavage, quantitation of DSB usage frequency by TrAEL-seq correlated well with S1-seq (R2=0.86) (Fig. 1F, left), and to a similar extent with Spo11- associated oligonucleotide sequencing (R2=0.84) (Fig. 1F, right)"

These correlations whilst useful are rather crude comparisons. Since an advertised benefit of the method is the nucleotide accuracy, a more detailed comparison showing how the nucleotide accuracy of S1-seq and TrAEL-seq compares within a zoomed-in strong hotspot region is expected. 

Indeed, since much is being made of the single-nucleotide accuracy, the authors should compare their dmc1 TrAEL-seq to meiotic CC-seq data (Gittens et al 2019 https://www.nature.com/articles/s41467-019-12802-5), which, unlike Spo11-oligo seq, has no mapping ambiguity (no tailing is involved).

Additionally, an important point about sensitivity is missing here: How many of the 3901 hotspots were detected by TrAEL-seq? How does this compare to the number detected by S1-seq (and/or CC-seq datasets)?

Specifically, the Y-axis of Fig 1F is about 5-fold lower in TrAEL-seq. Assuming all reads are expressed as hits per million mapped reads (or equivalent? *Please add this information to all figures*), then this strongly suggests that the signal to noise of TrAEL-seq is significantly lower than S1-seq, which itself is lower than Spo11-oligo seq.

Page 5. "Two RFB sites are readily visible in wild type TrAEL-seq data as peaks of antisense reads relative to the direction of replication" AND "At centromeres replicated from one direction only, we observed an accumulation of reads antisense to the direction of replication just before the centromere, while forks in termination zones that can be replicated in either direction displayed both peaks (Fig. 2D)."

These sentences are unclear. The use of "sense" and "antisense" is typically in relation to transcription, yet replication is mentioned instead. This section could be clarified by referring to the orientation of the 3' DNA end itself, relative to the leading/lagging strand. E.g. add: "…, corresponding to 3' DNA ends on the leading strand".

More specifically, do peaks on the R-strand indicate nascent (genomic) 3' ends on the F or R strands? Please revise text to make sure this information is absolutely clear throughout the manuscript.

Page 5. "although TrAEL-seq data displays higher signal-to-noise ratios than GLOE-seq data with peaks that correspond more closely to known sites than qDSB-seq

peaks (Fig. 2B) [13, 36]."

Where is the evidence for greater signal to noise in TrAEL-seq vs GLOE-seq? Perhaps the second replicate of GLOE-seq is worse...but what is the cause of this? Were TrAEL-seq replicates highly correlated? Where are these data demonstrating that fact (both genome-wide, and specifically at this locus?). Reproducibility of TrAEL-seq is a very important point that must be presented and commented upon.

Page 5. "It should be noted that the TrAEL-seq and the GLOE-seq datasets used for this

analysis derived from asynchronous cells whereas the cells for qDSB-seq had been tightly synchronised in S phase, underlining the high sensitivity of TrAEL-seq for stalled replication forks."

Please revise. Just because TrAEL-seq detects a signal in the asynchronous cells says nothing in of itself of the sensitivity compared to other assays (which is the impression teh text is trying to make). Specifically, the synchronized qDSB-seq signal is at least 20-fold stronger. If the authors want to compare the sensitivity, they should synchronise their cells to make the data comparable.

However...I am now additionally concerned that the authors have not normalised their data for total sequencing depth. This is essential if any comparisons are to be made between the various samples and sequencing techniques: All data need to be presented as reads per million mapped.

A table stating the read depths and mapping statistics of all individual libraries used here is also *essential*.

Page 5. "Importantly, no difference was observed between the rad52Δ signal and the wild type, showing that these double stranded ends are not normally processed by the homologous recombination machinery (Fig. 2B compare wild type and rad52Δ panels)."

As stated above, this comparison is only valid if read depths have been normalised between samples. Please clarify and revise the interpretation if not.

In general the logic of this section is very poorly introduced and explored. The authors make far too many implicit jumps in data interpretation without fist introducing the concepts that they are testing (i.e that if this signal is a reversed fork, one would expect Rad52 dependence). This needs radical revision to be acceptable text for a broad journal like PLoS Biology: in its current form, non-specialist (and even specialist) readers will be lost.

The fact that the peak is similar (assuming still true after normalisation) in the presence and absence of Rad52, needs more careful interpretation and clearer description. The jump to Holliday junction processing makes no sense without greater, slow, detail explaining the logic.

Page 5. "To determine the applicability of TrAEL-seq to mammalian cells, we generated TrAEL-seq libraries from 0.5 million human embryonic stem cells (hESC) that were either undifferentiated or subjected to retinoic acid-induced differentiation."

What is the purpose of the retinoic acid-induced differentiation here? Do the authors expect to see a difference at the rDNA? 

Furthermore, the data here is incredibly noisy, and far from convincing. How many other genomic sites shows such similar peaks? What methods do the authors have to demonstrate that these signals are real and not noise?

What was the total read depth of these human samples?

How is mapping accuracy validated within the R repeats? Can the spike be an artefact stemming from copy number differences between the sample and reference? Or due to mis-mapping of reads to the repeat array? All these details are completely glossed over, leaving this reviewer far from convinced by the data presented.

Page 5 and Fig 2DE. The three colours are incredibly hard to distinguish. Why are they not labelled in the plot itself? (why only in the figure legend?). Is the reader meant to interpret anything from the differences (or not) between these mutants?

I also have concerns that only smoothed data (I assume that is what the wiggly lines are?) is plotted rather than nucleotide-resolution peaks of the raw signals.

Fig 2E. The signals at the tRNA are intriguing, but lead to a query: Is there any possibility that nascent RNA can act as a 3' end for TrAEL-seq? From Fig 1A, I assume a 3' RNA end would efficiently ligate to the first adapter, and also be reverse-transcribed by the Bst2.0 polymerase. From the methods it appears that adapter 2 is 5' phosphorylated, thus would be able to ligate to the new DNA 3' end generated by Bst2.0 (even though it would be a putative a DNA/RNA hybrid molecule). This product would then be a substrate for PCR because a single DNA strand would have been created with adapter 1 on the left and adapter 2 on the right.

I would like the authors to respond to this and explain how they can exclude this as a molecule that they may be inadvertently detecting using TrAEL-seq. I recognise that the authors use RNAseT1 in their DNA preparations, but I do not know if this enzyme cleaves the DNA/RNA hybrids present at sites of nascent transcription that I am suggesting may a source of 3' RNA ends.

Separate form these comments above, where is the evidence that the peak in the tRNA gene is replication dependent (and thus a site of replication stalling?). The peak in the tRNA is similar in both local replication orientations. Could it instead be (if not a labelled RNA species), a site of increased DNA template breakage due to the high levels of transcription going on here?

Page 6. The section: "A replication signal of the same polarity was noted...yet up to 90% of TrAEL-seq reads emanate from the leading strand."

Is incredibly hard to follow in relation to the preceding text figures, as the authors now refer to the position of the 3' end (I think?) (rather than the antisense TrAEL-seq read). I would suggest that throughout the paper/figures, the inferred position of the 3' end is exclusively referred to, for clarity. This is how the data were presented in the GLOE-seq paper, which will also aid comparison by the reader.

Please clearly summarise/conclude this section: Is TrAEL-seq detecting nascent 3' ends of the leading strand? If so, please state this clearly! If this is the case, why are nascent 3' ends on the lagging strand not also detected? (there should be many more of them too).

Please also define RFD in the legend.

In general, it seems inappropriate to refer to the plot in Fig 3B as an RFD plot (really it is just a strand ratio), since at this point it is not clear what signal TrAEL-seq is detecting.

Page 6. "suggesting that double stranded ends are also formed during normal replication".

Please consider that ssDNA/dsDNA junctions are likely to be mechanically fragile (even in agarose), and may preferentially break during processing generating a structure labelled by END-seq.

Page 6. "…although with the opposite polarity as expected (Fig. 3F and S3D) [36]."

Incorrect figure references. I think the author is referring to Fig. 3G and S3E.

Page 7. "The TrAEL-seq profile of clb5Δ was very similar to wildtype across most of the genome, but certain origins were clearly absent or strongly repressed, resulting in extended tracts of DNA synthesis from adjacent origins (Fig. 4A, green arrows). This is as predicted for clb5Δ mutants and confirms that TrAEL-seq is indeed sensitive to changes in replication profile."

The clb5 data are convincing. Could the authors strengthen the manuscript further by adding a supplementary figure showing a global analysis of the differences between wild-type and clb5∆ RFD plots, relative to the differences between two samples not expected to have differing origin usage? E.g. two scatterplot analyses. 

Page 7. "stalling but not in the expected location."

Please clarify what the expected location was...the reader will be completely lost at this point without a more careful, and slower, presentation of ideas, observations, and interpretations.

Fig 4B. This figure is not very convincing. Why are only R-strand reads presented? PLease add an X- scale. What smoothing was applied to the data? (Was it a rectangle or a Hann window, please specify) What was the justification for such a smoothing window? Were the data normalised for relative read depth between the Raffinose and R+Galactose libraries? If the data in 4C are 100 bp bins

Page 8. "Together, these data show that replication profiling by TrAEL-seq is sufficiently sensitive to reveal subtle differences in fork direction and processivity."

Where is the evidence to indicate that what is detected by TrAEL-seq is "subtle"? By what benchmark are these subtle effects?

Page 8, discussion: "Here we have demonstrated that TrAEL-seq maps resected DSBs,"

Here and elsewhere in the text, I think it is important to clarify the text to make it clear that the TrAEL-seq technique maps the 3' end of resected DSBs. i.e. the presumed DSB end.

Figure 1. "Note that TrAEL-seq reads map antisense to the cleaved strand, reading the complementary sequence starting from the first nucleotide before the cleavage site."

This is unclear because the diagram depicts a DSB, where both strands are cleaved. Which is the "cleaved strand"? Could the figure be labelled more clearly to indicate the first sequenced base.

---

## [Decision Letter · Decision Letter 2]

10 Feb 2021

Dear Jon,

Thank you for submitting your revised Methods and Resources entitled "Genome-wide analysis of DNA replication and DNA double strand breaks by TrAEL-seq" for publication in PLOS Biology. I've now obtained advice from two of the original reviewers and have discussed their comments with the Academic Editor. 

Based on the reviews, we will probably accept this manuscript for publication, provided you satisfactorily address the remaining points raised by the reviewers. Please also make sure to address the following data and other policy-related requests.

IMPORTANT:

a) Please attend to the remaining requests from the reviewers.

b) I wonder if the title might be slightly easier to parse if you use "using" instead of "by." Or flip it round to give "TrAEL-seq is a method for genome-wide analysis of DNA replication and DNA double-strand breaks"...

c) Please address my Data Policy requests further down.

We expect to receive your revised manuscript within two weeks. 

*Published Peer Review History*

*Early Version*

Best wishes,

Roli

Senior Editor,

rroberts@plos.org,

PLOS Biology

DATA POLICY:

Regardless of the method selected, please ensure that you provide the individual numerical values that underlie the summary data displayed in the following figure panels as they are essential for readers to assess your analysis and to reproduce it: Figs 1CDEF, 2BCDE, 3ABCDEFGH, 4ABCDEF, S1CDEF, S2ABCDEF, S3ABCDEFGH, S4ABCDE. NOTE: the numerical data provided should include all replicates AND the way in which the plotted mean and errors were derived (it should not present only the mean/average values).

REVIEWERS' COMMENTS:

Reviewer #1:

[identifies himself as Conrad Nieduszynski]

I would like to commend the authors on the excellent work they have undertaken to address reviewer comments. The manuscript is much improved and I believe ready for publication. 

Minor comment: "although TrAEL-seq data contains less additional peaks in this region than" should be "fewer" rather than "less".

In the revised manuscript the authors make the observation of phasing between TrAEL-seq signal and nucleosomes. This is precisely what I was anticipating seeing. Replication fork velocity is stochastic (just compare the length of tracks seen in multiple double pulse labelled combing/fibre studies), but the source of this variability is unknown. Multiple lines of evidence suggest this is sequence independent. Therefore, a possible explanation is that DNA replication frequently 'pauses', if only very briefly, as the fork pushes through nucleosomes (and as new Okazaki fragments are primed). I suggest that this would provide an explanation for the authors observation on phasing. Following this up is far beyond the scope of this manuscript, but in the future should be possible with various mutants that alter the position of nucleosomes and/or the ability of the fork to replicate through nucleosomes.

Reviewer #3:

TRAEL-seq second round review

The authors have done an excellent job answering the extensive comments and suggestions that were provided during the review process. The rebuttal comments are measured and well explained. The updated text and figures have improved clarity. I thank the authors for the time spent providing additional analysis, comparison to other methodological techniques, and for providing a detailed breakdown of the datasets presented in the study (in particular to include statistics on mapping and de-duplication). My main concern with data presentation was clarity on which strand was being referred to, and normalisation of all data to reads per million mapped. Both changes have been incorporated. The authors additionally provide G1-arrested, and even a very exciting attempt at the G1 > S release. Both sets demonstrate clearly the S-phase dependence of the mapped signals.

I have one minor comment/suggestion that I think will aid a non-specialist reader. Would it be possible to make the diagrams of the rDNA in Fig S3B and Fig 2B more consistent with one another? Specifically, how do the three RFBs referred to in Fig 3B (and the two strong TRAEL-seq peaks) relate to the single RFB (and dispersed TRAEL-seq peak profiles) in Fig S3B? Why do the data look so different?

Also, it is not stated, but is the orientation of the rDNA repeat drawn consistent with the orientation in the standard S. cerevisiae reference sequence? This is probably obvious to the authors, but since there are no chromosomal coordinates indicated in Fig 3B, it is not clear, and thus perhaps this information can be added to the legend.

Overall, this is an interesting and potentially very useful analysis method that expands the tools available to researchers in a new and sensitive way. The authors should be commended on its development, and on their manuscript.

---

## [Editor Report · Decision Letter 3]

17 Feb 2021

Dear Jon,

On behalf of my colleagues and the Academic Editor, Tanya Paull, I'm pleased to say that we can in principle offer to publish your Methods and Resources paper "Genome-wide analysis of DNA replication and DNA double strand breaks using TrAEL-seq" in PLOS Biology, provided you address any remaining formatting and reporting issues. These will be detailed in an email that will follow this letter and that you will usually receive within 2-3 business days, during which time no action is required from you. Please note that we will not be able to formally accept your manuscript and schedule it for publication until you have made the required changes.

PRESS: We frequently collaborate with press offices. If your institution or institutions have a press office, please notify them about your upcoming paper at this point, to enable them to help maximise its impact. If the press office is planning to promote your findings, we would be grateful if they could coordinate with biologypress@plos.org. If you have not yet opted out of the early version process, we ask that you notify us immediately of any press plans so that we may do so on your behalf.

Thank you again for supporting Open Access publishing. We look forward to publishing your paper in PLOS Biology. 

Best wishes,

Roli 

Roland G Roberts, PhD 

Senior Editor 

PLOS Biology